

# Simulating multivariate hazards with generative deep learning

Alison Peard[1], Yu Mo[1], and Jim W. Hall[1]

[1]Environmental Change Institute, University of Oxford, Oxford, UK

**Correspondence:** Alison Peard (alison.peard@ouce.ox.ac.uk)

**Abstract.** When natural hazards coincide or spread over large areas they can create major disasters. For accurate risk analysis, it is necessary to simulate many spatially resolved hazard events that capture the relationships between extreme variables, but this has proved challenging for conventional statistical methods. In this article, we show that deep generative models offer a powerful alternative method for creating sets of synthetic hazard events due to their ability to implicitly learn the joint

distribution of high-dimensional data. Our framework combines generative adversarial networks with extreme value theory to construct a hybrid method that captures complex dependence structures in gridded multivariate weather data and provides a theoretical justification for extrapolation to new extremes. We apply our method to model the co-occurrence of strong winds, low pressure, and heavy precipitation during storms in the Bay of Bengal, demonstrating that our model learns the spatial and multivariate extremal dependence structures of the underlying data and captures the distribution of storm severities. Validation

shows excellent preservation of spatial correlation structures (r = 0.977, MAE = 0.053) and multivariate dependencies (r = 0.817, MAE = 0.096) for wind, precipitation, and pressure fields. In a case study of storm risk to mangrove forests, we demonstrate that correctly modelling the dependence structures leads to far more realistic estimates of aggregate damages. While our method shows mild underestimation of the damages with a mean absolute error of 93.57 $\text{km}^2$, this remains an order of magnitude lower than errors from independence assumptions (460.54 $\text{km}^2$) and the total dependence assumption (1056.90

$\text{km}^2$) that is implicit when using return period maps. The framework developed in this paper is flexible and applicable across a wide range of data regimes and hazard types.





# 1 Introduction

## 1.1 Background

Multivariate and spatially compounding hazards can lead to disasters of tremendous severity and complexity (Zscheischler
et al., 2020). Multivariate compound hazards occur when different hazards coincide over the same region. For example,
droughts often coincide with heatwaves, while strong winds frequently accompany heavy rainfall (Yin et al., 2023). During Hurricane Sandy of 2012, disastrous inland flooding was driven by the combined effects of rainfall, strong winds, tides,
and storm surges (Zscheischler et al., 2018). Furthermore, the vulnerability of natural and man-made assets typically exhibits
nonlinear relationships with hazards. Damage to mangroves during storms, for instance, results from the cumulative effects of
rainfall, wind speed, and storm speed, alongside other climatological, biological, and geophysical variables (Mo et al., 2023;
Taillie et al., 2020; Amaral et al., 2023).

Spatially compounding hazards occur when climate hazards spread over large regions and create strain on distributed systems. Examples of this include droughts hitting multiple global breadbaskets and creating food shortages, or widespread storm
damages stretching emergency response capacity (Zscheischler et al., 2018; Boulaguiem et al., 2022; Mehrabi, 2020; Gaupp
et al., 2019). In 2002, droughts hit Europe, Russia, India, and China, leading to significant decreases in the global production
of rice and wheat (Bailey et al., 2015). The risk of simultaneous breadbasket failures is estimated to increase with global mean
temperature, rising from 6% to 40% for maize between historical (0.85°C) and 1.5°C levels (Gaupp et al., 2019).

Hazards and their drivers exhibit highly variable levels of dependence across space and between variables. In spite of this
heterogeneous dependence, climate risk to human or natural systems is often assessed by examining hazard variables individ-
ually at a specific location or by treating hazard maps like events. This implicitly assumes total dependence across space. A
1-in-50 year flood map, for example, represents a flood with a 50-year return level depth in all locations simultaneously. In
reality, however, we expect such a homogeneously distributed flood to be far rarer than a 1-in-50 year event. Such simplifica-
tions omit the effects of spatially compounding hazards and generate significant bias (Lamb et al., 2010; Metin et al., 2020).;
however, assumptions such as these have been employed in high-profile studies. In an article that informed a flagship World
Bank report, Maruyama Rentschler et al. (2019) treated flood hazard maps like real events and used them in a transport routing
analysis for major cities in Mali, Tanzania, Uganda, and Rwanda. Such an assumption may be justified at a city scale, but in
this paper we will demonstrate the bias created when employing such assumptions on larger national or continental scales.

A second limitation of hazard maps is that they are univariate and do not account for multi-hazard risk. Thus, any climate
risk analysis relying on hazard maps necessarily treats different hazard types independently. Such an approach risks neglect-
ing some of the most potentially disastrous climate risks. How likely is it for high temperatures and drought to hit multiple
continents simultaneously? What is the probability of multiple villages along a coastline being hit by storm surge such that
emergency services are at capacity? How likely is it that this storm coincides with high tides or pluvial flooding? With univariate
approaches, the probability of such disastrous events remains unquantified.

As climate change shifts the distribution of climate variables, the probability of co-occurring extremes may also increase
(Mehrabi, 2020). The spatially compounding problem has motivated many risk analysts to move towards an event-based ap-



proach, estimating the damages across sets of thousands of plausible hazard events and producing statistics for the resulting damages (Bloemendaal et al., 2020; Cross et al., 2020; Bates et al., 2023b). Synthetic event sets are widely used for catastrophe modelling in the insurance industry and have been used for several large-scale infrastructure risk assessments in the UK (Lamb et al., 2010; Bates et al., 2023a).

For simulating synthetic hazard event sets, statistical methods have been widely used (see Sect. 1.2) yet have proven to be limited in their capacity to accurately simulate multivariate extreme events over large domains. In this article, we harness a deep learning framework to produce spatially coherent multi-hazard event ensembles. These event ensembles should preserve the statistical dependence properties of historical climate data in a spatially coherent manner, especially in the extremes. Our criteria for success are that the generated data: (i) replicates the overall distribution of extreme events; (ii) preserves the

dependence structure of the training data across space—in the bulk and in the tails; (iii) preserves the dependence structure between different variables—in the bulk and in the tails; (iv) preserves the marginal distributions of the training data; and (v) has a well-justified basis for extrapolation to new, more extreme events.

## 1.2   Statistical approaches

There is a rich—though at times much-debated (Mikosch (2006); Embrechts (2006); Harris (2005))—literature concerning the

statistics of univariate and multivariate extremes (Coles et al., 2001; Heffernan and Tawn, 2004; Cooley et al., 2012; Davison and Huser, 2015), with a dense sub-literature of applications to hydrological and climate data (Brunner et al., 2021; Lamb et al., 2010; Schoelzel and Friederichs, 2008). To calculate statistics over possible damages, rigorous climate risk analysis requires samples of thousands of extremes from the joint distribution of potentially compounding climate hazards. Two popular approaches to estimate the joint distribution are copula methods and conditional exceedence models (Nelson, 2006; Davison

et al., 2012; Heffernan and Tawn, 2004). Both approaches decompose the problem into modelling the univariate marginal distributions and learning the dependence structure between the marginals, usually after some probability space transformation. Copulas are based on Sklar's theorem and accurate modelling hinges on choosing the most appropriate copula family for the dependence structure (Sklar, 1959). The conditional exceedence model of Heffernan and Tawn (2004) calculates the probability that each variable is extreme, given that some conditioning variable is already extreme. This introduces flexibility into the

asymptotic dependence structure between bivariate pairs.

    Max-stable processes have also gained popularity in practical application since Padoan et al. (2010)'s development of likelihood-based inference methods, and can be loosely interpreted as an extreme extension of Gaussian random fields for annual maxima (Huser et al., 2024). They suffer a number of limitations, however, which are reviewed in-depth in Huser et al. (2024). In particular, annual maxima almost never represent a real event over large regions; the max-stability requirement

implies that the spatial dependence structure is independent of block size; sub-asymptotic tail dependence is not captured; and most approaches to fitting the likelihood function become computationally expensive as the number of locations modelled increases.

    For the specific case of tropical cyclones, various stochastic–empirical methods have been used to simulate realistic cyclone tracks and intensities for risk analysis, mostly based on IBTrACS (Knapp et al., 2010; Bloemendaal et al., 2020; Lee et al.,





2018; Sparks and Toumi, 2024). The STORM dataset of Bloemendaal et al. (2020), for example, follows a stochastic-empirical approach and models wind speed using an empirical relationship for pressure, whose evolution is predicted using a constrained autoregressive process. The autoregressive process is not an extremal model, however, which somewhat limits the justification for extrapolation to unseen extremes.

## 1.3 Data-driven approaches

Data-driven approaches to modelling multivariate extremes in high dimensional spaces have been evolving over the last two decades (Heffernan and Tawn, 2004; de Fondeville and Davison, 2018; Cooley and Thibaud, 2019; de Fondeville and Davison, 2022; Engelke and Ivanovs, 2021; Wadsworth and Tawn, 2022) but, with the recent strides made in generative artificial intelligence (Bond-Taylor et al., 2021), data-driven methods are being catapulted to the forefront of research agendas. Since 2021, several innovative empirical methods, which we outline here, have been developed to address the statistical modelling of high 95 dimensional multivariate extremes.

Engelke and Ivanovs (2021) give an overview of the limitations of classical statistical methods for representing extremes in high-dimensional settings and propose two unsupervised learning approaches, clustering and principal component analysis, to reduce the dimensionality of the problem. Li et al. (2023) use k-means clustering to cluster a global dataset of extreme sea levels and combine this with a conditional exceedence method to generate a global event set. Wilson et al. (2022), in a 100 model called DeepGPD, use fully connected and convolutional neural networks to predict the shape and scale parameters of generalized Pareto-distributed asymptotics above a certain threshold, regressing against exceedences and other covariates from the previous time step.

Many authors leverage a deep learning model known as a generative adversarial network (GAN) which has several appealing properties, including the ability to implicitly learn the distribution of the training data (Mohamed and Lakshminarayanan, 105 2016). GANs consist of two deep neural networks: a generator and a discriminator. The networks are trained adversarially: the generator maps a random latent space to an image space, and the discriminator must distinguish between real and generated images. It has been shown, however, that GANs tend to produce samples near the bulk of the distribution (Huster et al., 2021). For this reason, several methods have emerged to adapt GANs to produce more extreme samples, in controlled and theoretically justified ways. We recommend Prince (2023) for a good introduction to the fundamentals of GANs.

Bhatia et al. (2021) propose *ExGAN*, which uses extreme value theory to assign a measure of extremeness to cumulative precipitation over the contiguous United States. They employ a data-shifting routing to filter and shift the data distribution towards the extremes, and use this to generate samples conditional on desired extremeness. Huster et al. (2021) prove that, provided it has unbounded activation functions, the tails of data generated by a GAN will have the same shape as those of underlying latent space, which is usually Gaussian. They propose *ParetoGAN*, which uses a generalised Pareto latent space 115 and a root-Euclidean energy distance loss function, and demonstrate on synthetic datasets that this better captures power-law behaviour.

Boulaguiem et al. (2022) fit generalized extreme value (GEV) distributions to a spatial grid of annual maximum temperature and precipitation over western Europe and perform a probability integral transform (PIT) before training a deep convolutional





GAN (DCGAN) on the images with uniform marginals (Radford et al., 2015). They use the fitted GEV parameters to transform data back to the original scale and guide extrapolation beyond the range of the data. Their model *evtGAN* is flexible and can learn complex nonlinear and spatially nonstationary dependence structures. It can also allow for mixtures of heavy and light-tailed marginal distributions. The authors demonstrate that incorporating the PIT step significantly improves the representation of extreme samples versus a DCGAN, whilst maintaining good representation in the bulk of the data. However, evtGAN is trained on annual block maxima, so does not account for compound hazards on smaller time scales, such as storms occurring over a few days. It is also univariate, so it does not cover multivariate compound events.

Our framework builds on the work of Boulaguiem et al. (2022) to increase temporal granularity and model multivariate events, thus creating a powerful tool for climate risk analysis of large-scale systems.

## 1.4 Our approach

This article proposes *hazGAN*, a modelling framework that builds upon the work summarized in Sect. 1.3 to create datasets which are directly applicable in risk analysis. We begin with the model of Boulaguiem et al. (2022) and make the following series of modifications:

- We replace the annual block maxima approach with a peak-over-threshold (POT) approach, allowing us to use more training data, limit parametric assumption-making to the tails, and draw stronger conclusions about the degree of dependence across space and between variables.

- The POT approach allows us to work with hazard *footprints*, which are already commonly used in risk modelling and have a trivial extension to multivariate events (Steptoe and Economou, 2021). We generate hazard footprints from gridded climate data using a runs declustering approach (Coles et al., 2001).

- We train GANs with multiple channels, such that each channel represents a different climate variable, and thus images represent multivariate hazard footprints.

- We transform images such that all marginal distributions along the time dimension are Gumbel-distributed, and train the GAN on these, which encourages it to be sensitive to dynamics in the extremes.

We demonstrate our method's effectiveness by training the model on historical footprints of wind speed, precipitation, and sea level pressure corresponding to wind storms over the Bay of Bengal. We then use these to construct risk profiles for the total area of storm damage to mangrove forests across the region. We choose mangroves as an illustrative example because damage to mangrove forests during storms is driven by a combination of rainfall and wind speed (Mo et al., 2023; Taillie et al., 2020; Amaral et al., 2023). This multivariate vulnerability makes it difficult to conduct comprehensive scenario modelling of risk to mangrove forests, especially over large areas. Despite the increasing recognition of the valuable ecosystem services provided by mangroves (Menéndez et al., 2020; Duarte et al., 2013), and the documented loss of mangroves over the last decade (Goldberg et al., 2020), comprehensive large-scale scenario modelling of wind storm risk to mangrove forests has not been attempted to date.



We present results from our illustrative example and show that our model learns spatial and multivariate dependence structures over large regions, even in the extremes. We additionally demonstrate that our method produces far more realistic estimates of annual mangrove damages, compared to the total dependence assumption which underlies hazard maps, or assumptions of total independence between variables and space, which are implicit in single-location studies. Thus, we replicate the
spatial dependence scenarios assessed for flooding in Metin et al. (2020).

To the best of the authors' knowledge, this framework is the first such framework to facilitate multivariate extreme event set generation over such large scales. It is flexible and can be used for any variables for which there exists suitable data. Though applied to gridded reanalysis data in this paper, the method can be used to augment any gridded dataset. Application to global climate model data instead of reanalysis data, for example, would follow an identical method. By using different deep learning
architectures, the method could also be extended to different data topologies, and be applied to networks of river gauges or weather stations.

The remainder of this article is structured as follows: Section 2 reviews concepts in extreme value theory and deep learning fundamental to our approach. Section 3 outlines the method. Section 4 applies the method to a concrete example of wind storms in the Bay of Bengal and evaluates its performance. Section 5 demonstrates a use case for the synthetic event set, analysing
storm risk to mangrove forests in the Bay of Bengal. Finally, Sections 6 and 7 summarize the contributions of this paper and outline potential future directions for the research.



## 2 Theory

### 2.1 Extreme value statistics

We are particularly concerned with the behaviour of climate variables in the extremes, for example, the strongest winds,
heaviest precipitation, or lowest sea-level pressure. For risk analysis, we want to project to values more extreme than previously
observed and it is important for a trustworthy model to underpin this extrapolation. As such, the choice of model must be well-
justified, either by mathematical result or sufficient empirical evidence. In extreme value statistics, asymptotic models are a
popular choice as they have rigorous mathematical foundations (Huser et al., 2024). The Fisher–Tippett–Gndenko theorem
proves that, under certain conditions, the limiting distribution for block maxima—as block size approaches infinity—is the
generalized extreme value (GEV) distribution. Similarly, the Pickands–Balkema–de Haan theorem proves that the limiting
distribution for exceedences over a threshold approaches the Generalized Pareto (GPD) distribution as the threshold approaches
infinity. Despite their strong theoretical foundations, however, these models suffer from a bias–variance trade-off, particularly
in low-data regimes. The threshold or block size must be high enough that it is appropriate to employ the asymptotic results,
but there must still be enough data to limit variance in the fit.

Furthermore, for many climate variables, there is strong empirical evidence for the existence of certain parent distributions,
with known asymptotic forms. For wind speed in particular, a wealth of evidence supports the existence of a Weibull parent
distribution, which some practitioners refer to as a "stretched exponential" type distribution because it stretches out the tails
of the exponential decay. This has been observed for temperate storms, thunderstorms, and tropical cyclones, as well as mixed
climates (Harris, 2005).

According to the Pickands–Balkema–de Haan theorem, as the threshold increases, distributions of the stretched exponential
family converge to their limiting distributions extremely slowly. Hence a prohibitively large volume of data is required before
it is appropriate to apply an asymptotic fit (Papalexiou et al., 2013). Additionally, the limiting distribution for a Weibull parent
distribution is the Gumbel distribution (Type I GEV with zero shape parameter). Harris (2005) notes that neither GPD nor
GEV methods can produce a zero shape parameter as it corresponds to a singularity in the likelihood function. Fréchet fits
are rare, so attempting to fit a GPD or GEV to Weibull parent data usually produces a Type III fit, which corresponds to
a negative shape parameter and upper bound and is sometimes referred to as a *reversed Weibull* distribution. This has been
considered acceptable due to the belief that wind speeds have a natural upper bound; however, Harris (2005) writes that
this is not sufficient evidence for the adoption of a Type III fit and cautions that any constraint on maximum wind speeds
must be rigorously justified, especially in risk analysis. Consequentially, for this work, we will assume either empirically or
asymptotically justified distributions to model climate variable extremes, depending on existing evidence in the literature.

#### 2.1.1 Semiparametric distribution function

To make the best use of small (fewer than 100 years of) climate datasets, we will choose a peak-over-threshold approach for
fitting distributions to climate variables and follow the semiparametric approach used by Heffernan and Tawn (2004) in their
conditional exceedence model. However, we will use a generalization that allows for arbitrary parametric distributions to be





fitted above the threshold, rather than only that of the GPD distribution. This allows us to use different distributions to model the exceedances in cases where this is deemed more appropriate.

For a random variable $X$ and suitably extreme threshold $v_X$, the three-parameter semiparametric distribution is given by,

$$
\tilde{F}(x) = \begin{cases} 1 - (1 - \hat{F}(v_X))(1 - F_{\xi,\mu,\sigma}(x)) & \text{for } x > v_X \\ \hat{F}(x) & \text{for } x \leq v_X \end{cases}
\tag{1}
$$

where $\hat{F}$ is the empirical distribution function (ECDF) and $\xi$, $\mu$, and $\sigma$ are the shape, location, and scale parameters. The meaning of these parameters depends on the distribution in question, but they can be generally understood to describe the tail behaviour, lower bound, and spread of the random variable, respectively.

For any random variable $X$, it is well-known that $F(X)$ is uniformly distributed: $F(X) = U \sim \mathcal{U}(0,1)$. This fact can be used to transform a random variable to any arbitrary distribution $\mathcal{D}$. Provided the distribution function $F_{\mathcal{P}}$ is known, $F_{\mathcal{P}}^{-1}(U) \sim \mathcal{D}$.
This very useful transform is known as the probability integral transform (PIT) and we will make extensive use of it in this work.

### 2.1.2 Extremal dependence metrics

Simply extrapolating every variable to new extremes is insufficient, it is also necessary to capture the dependence between multiple variables in the extremes. If two climate variables are likely to reach extreme values simultaneously, this has im-
portant implications in risk analysis. We require extremal dependence metrics to measure and compare the level of extremal dependence between different variables.

Standard dependence measures such as correlation are limited because variables that are correlated near their means won't necessarily exhibit dependence in their extremes. A more specialized metric, the *extremal coefficient* $\theta$, provides a measure of the extremal dependence between any number of variables. For $K$ random variables, $\theta$ takes values in $[1, K]$. This has
an intuitive interpretation as the effective number of independent variables present, i.e., $\theta = 1$ implies that effectively only one independent variable is present or that the variables are totally dependent—in the extremes. We can construct the extremal correlation, an analogue to the Pearson correlation, as $\chi = (K - \theta)/(K - 1) \in [0,1]$. In an unpublished but much-cited manuscript, Smith (1990) derived an estimator for the extremal coefficient given $N$ samples and $K = 2$ Fréchet-distributed random variables $Y_1, Y_2 \sim \mathcal{F}$,

$$
\hat{\theta}_{12} = \frac{N}{\sum_{n=1}^{N} \min(y_{n1}^{-1}, y_{n2}^{-1})}
\tag{2}
$$

which the author calls the "raw estimates" of the extremal coefficient, with a natural extension into $K$ greater than 2. The estimator $\hat{\theta}$ can take values in $[1, \infty)$ and $\hat{\theta} > K$ corresponds to negative extremal dependence. The Smith estimator is quite sensitive; however, and can produce values outside the theoretical bounds $[1, K]$, particularly when applied to finite samples or samples that deviate from the Fréchet assumption.



A second approach uses the *tail dependence coefficient* $\lambda$ (Joe, 1997), which directly measures the conditional probability of joint extreme events. For two variables with marginal distributions $F_X$ and $F_Y$, the upper tail dependence coefficient is defined as

$$\lambda_u = \lim_{u \to 1^-} P(F_Y(Y) > u \mid F_X(X) > u) \tag{3}$$

where $\lambda_u \in [0, 1]$. This coefficient has a natural interpretation: $\lambda_u = 0.3$ means that where one variable exceeds its 90th percentile, there is a 30% probability the other variable also exceeds its 90th percentile. Unlike extremal coefficients, tail dependence coefficients are naturally bounded, making them more robust to practical applications (Nelson, 2006).

## 2.2 Generative adversarial networks

Generative adversarial networks (GANs) were introduced by Goodfellow et al. (2014) and a large literature of theoretical advancements, extensions, and modifications has been growing ever since (Bond-Taylor et al., 2021). The original GAN places two neural networks in competing roles. A data generator—which never sees the training data—produces samples, and the critic—which sees a jumble of real and generated data—guesses which are real and fake. The networks are trained in alternating fashion. In this way, the generator learns to reproduce the distribution of the training data and generate datasets indistinguishable from the training data in appearance and distribution. Classic GANs require tens of thousands of training samples to prevent generator overfitting and stabilize training (Karras et al., 2020; Zhao et al., 2020). Even then, the training dynamic is extremely sensitive to hyperparameter choice and relies on neither model overpowering the other (Lucic et al., 2018). Numerous extensions, such as the introduction of Wasserstein loss functions and gradient penalties have improved this situation (Arjovsky et al., 2017; Gulrajani et al., 2017), but ultimately the attention of the scientific community turned towards newer, more stable generative models such as diffusion models (Yang et al., 2024). However, almost a decade of scientific research has resulted in a diverse and well-documented set of GANs. The StyleGAN series of GANs from nVIDIA remains the state of the art. These models have been extensively tuned to require significantly less data (as little as 1,000 samples), and further modifications to them have reduced this number to only 100 samples (Karras et al., 2020; Zhao et al., 2020).

## 2.3 Event footprints

Event footprints are a useful tool in catastrophe modelling, capturing the maximum intensity of a hazard event over its lifetime for each point in space (Steptoe and Economou, 2021). These footprints are created by aggregating hazard-related variables over time at each location. The method for defining peak hazard varies by climate variable—some require accumulation over time, while others use statistical measures such as maximum or minimum values.





## 3 Method

This section outlines the general methodology of hazGAN, which can be divided into three phases: (i) event footprint extraction, (ii) transformation of the marginal distributions, and (iii) GAN training and inference. The method is flexible and can be applied to different gridded datasets, climate variables, and locations.

Only four inputs need to be changed for new studies: (i) a region of interest must be defined by its bounding box; (ii) data cubes with dimensions $H \times W \times T$ must be supplied for each climate variable; (ii) a temporal aggregation function $h_{k|t}(\mathbf{x})$ along the time dimension $(t)$ must be defined for each climate variable; and (iv) $r_{|ijk}(\mathbf{x})$ a hazard definition function or *risk functional* must be defined across all space and variables $(ijk)$.

Throughout this paper, latitude will be indexed by $i = 1, \ldots, H$; longitude by $j = 1 \ldots W$; time by $t = 1, \ldots T$; sample number by $n = 1, \ldots, N$; and climate variables by $k = 1, \ldots, 3$. The notation $_{|ijkt}$ will be used to indicate which dimensions of a tensor a function is applied over. Data cubes with their original marginal distributions are denoted by $\mathbf{x}$, with uniform marginals by $\mathbf{u}$, and with marginals transformed to any other distribution $\mathbf{y}$.

### 3.1 Event footprints

The first step extracts multivariate hazard footprints from the climate data cubes. Figure 1 gives a schematic of the event footprint extraction stage. For each climate field, a spatiotemporal data cube for the region of interest must be supplied. If observations are sub-daily, $h_{k|t}(\mathbf{x})$ is applied along the time dimension to resample daily observations. Next, a deseasonalization function $s_{|t}(\mathbf{x})$ is applied along the time dimension to remove seasonality from each variable.

The deseasonalised data cubes are transformed into a time series using a *risk functional* $r_{|ijk}(\mathbf{x})$ (de Fondeville and Davison, 2022). This function determines what types of hazards are extracted, e.g. calculating the mean or sum of a variable will prioritize widespread events while a maximum or minimum will identify more localized events that reach higher peak intensities. de Fondeville and Davison (2022) give an in-depth discussion on the effects of different $r_{|ijk}$ choices and the choice of risk functional in our application is discussed further in Sect. 4.1.1.

A declustering algorithm $d_{|t}(\cdot)$ is applied to the $r_{|ijk}(\mathbf{x})$ time series to identify hazard events. Hazard days are consecutive days in which $r_{|ijk}(\mathbf{x})$ exceeds some threshold $v_r$, separated by some number of days of non-exceedences $\ell_r$ (Coles et al., 2001). Finally, the deseasonalised data cubes are intersected with the hazard days to create a hazard event set, which is then collapsed along the time dimension using $h_{|t}(\mathbf{x})$ to generate a set of event footprints.



**Parameters**

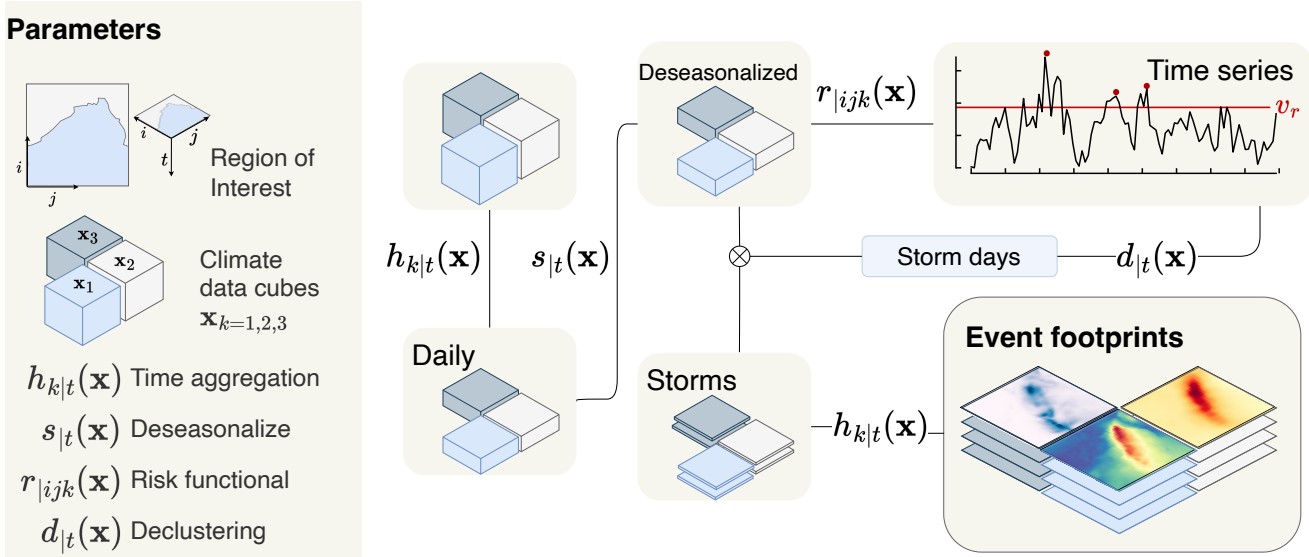

**Figure 1.** Schematic of the workflow to extract hazard event footprints. Gridded hourly climate data over a region of interest for three variables (indexed $k = 1, 2, 3$) is resampled to daily aggregates according to resampling function $h_{k|t}(\mathbf{x}_k)$. The daily aggregates are deseasonalised according to $s_{|t}(\mathbf{x})$. A risk functional $r_{|ijk} : \mathbb{R}^{T \times H \times W \times K} \to \mathbb{R}^T$ constructs a time series from the deseasonalised data and a declustering algorithm $d_{|t}(r_{|ijk}(\mathbf{x}))$ identifies storm days. Data cubes of daily aggregates are extracted for each storm using the storm days and these are aggregated into storm footprints by again applying the $h_{k|t}$ functions along the time dimension.

In order to fit parametric models to the marginals, the footprint data must be independent along the time dimension, which favours using high thresholds and extracting fewer storms. However, this will not facilitate fitting a distribution to the extremes

of each marginal, especially if storms occur in different regions at different times. In other words, one set of days may have lots of extreme values for one location but not capture any of the extremes in another location. Additionally, having more data is important for deep learning. To address this, we can reduce the threshold on $r_{|ijk}\mathbf{x}$) at which storms are defined and extract more storms. In this work, the choice of threshold is optimized using a grid search to identify the largest number of storms possible, while maintaining independence between clusters. To assert storm independence, we only require that the

peak $r_{|ijk}(\mathbf{x})$ values are independent between each storm and we use a Ljung–Box test to check this (Ljung and Box, 1978).

### 3.2 Marginal transforms

Figure 2 shows the marginal transformation workflow which takes the set of event footprints from Sect. 3.1 as input. The marginal distributions of the grid cells are defined along the time dimension, so there are $H \times W \times K$ marginal distributions to be fitted. For each marginal, the semiparametric distribution function in Eq. (2.1.1) is used to transform it into a uniform

distribution. The parameters of each marginal are fitted using maximum likelihood estimation and tested for goodness-of-fit





using an Anderson–Darling test. If significant $p$-values are obtained repeatedly, an empirical distribution function is used as a fallback. Suitable thresholds are selected using the transformed $p$–values method of Bader et al. (2018).

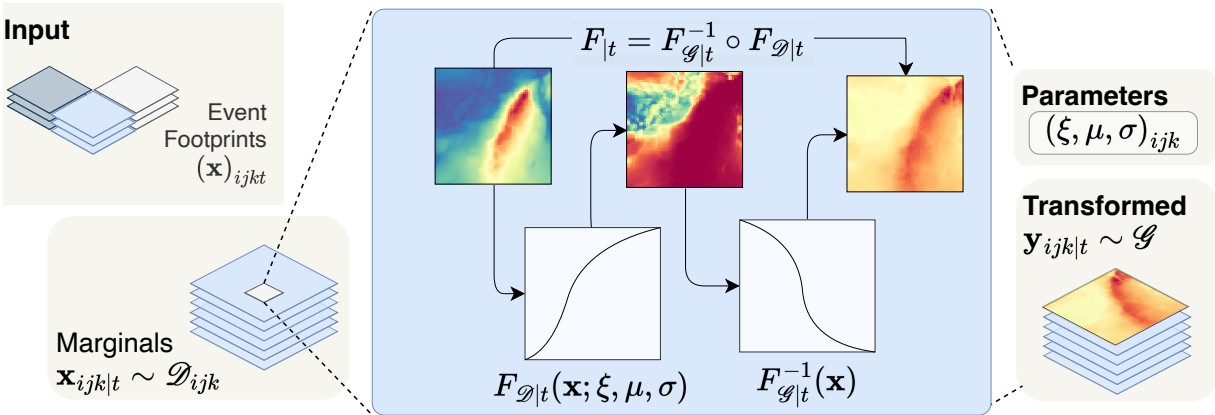

**Figure 2.** Schematic of the workflow to transform the marginal distributions of the storm footprints from Fig. 1 to a standard Gumbel distribution. A suitable parametric distribution is fitted to the extremes of each climate variable along the sample (time) dimension for each variable and location (marginal) $x_{ijk|t} \sim \mathcal{D}_{ijk}$. The semiparametric distribution Eq. (2.1.1) transforms each marginal to a standard uniform distribution, using the fitted parameters. The Gumbel quantile function then transforms each uniformly distributed marginal to a standard Gumbel distribution. The output set of footprints $\mathbf{y}$ have a standard Gumbel distribution along the sample (time) dimension.

At this point, all the marginals have the same standard uniform distribution. This makes them appropriate for training a neural network, which requires all data to be on the same scale. The uniform distribution, however, places no particular emphasis on extremal values and since we are interested in the tails of the data, we want the network to be very sensitive to variation in high ($\geq$90th) quantiles. Transforming to a Gumbel distribution $\mathcal{G}$, using

$$\mathbf{y} = -\log(-\log(\mathbf{x}))$$

will stretch out high values of a random variable and encourage the network to pay more attention to data in this range. Thus, we transform all the variables to Gumbel distributions.

## 3.3 Training and sampling

The footprints are now almost ready to be used for generative model training. However, the Gumbel distribution has an unbounded domain that lies mostly in $(-2, 13)$ and the footprints need to be rescaled to $[0, 1]$ before they are ready for training. This can be done using simple max-min scaling. After rescaling, the climate fields are stacked such that each multivariate footprint is now a three-channel tensor. Each footprint is converted to an RGB image, ready to feed into a generative model. A generative model is trained on the images and used to generate thousands of synthetic RGB images with the same distribution. We invert the scaling on these to recover Gumbel-distributed marginal distributions, apply the Gumbel quantile function to





obtain uniform samples, and apply the inverse of Eq. (2.1.1) to get synthetic data on the original scale. Depending on the deseasonalization function $s_{|t}(\mathbf{x})$ used, the inverse of this may also be applied. The result at this point is a large multivariate hazard event set ready for risk analysis applications.

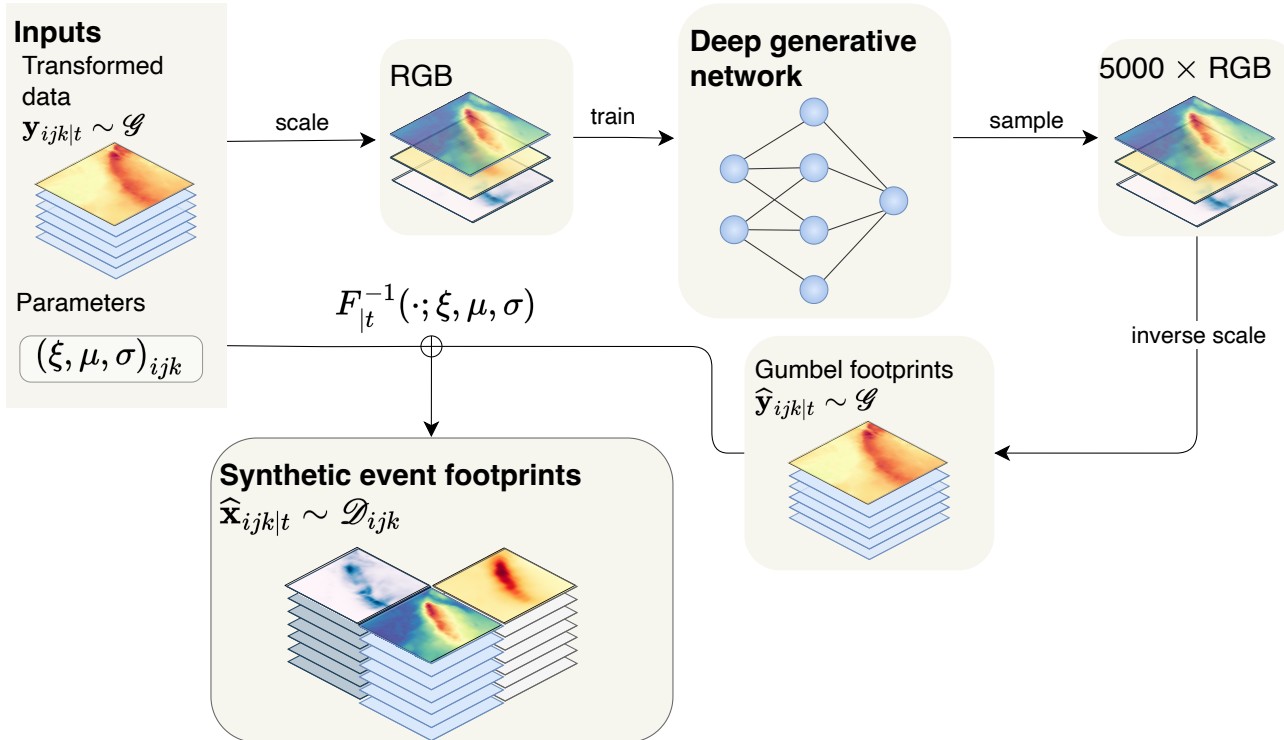

**Figure 3.** Schematic of the workflow for training the deep generative model. The Gumbel-distributed storm footprints from Fig. 2 are rescaled to take values in the range $(0, 1)$ using min-max scaling and are converted to three-channel RGB images. A deep generative network is trained on these images. To create synthetic storm footprints, thousands of new samples are generated from the generative model and these are re-scaled to have standard Gumbel marginal distributions using inverse min-max scaling. The inverse of the probability integral transform described in Fig. 2 is used to convert the synthetic footprints back to the scale of the original (deseasonalised) data.



## 4 Application: Wind storms in the Bay of Bengal

This section presents an illustrative example of our hazard simulation method by generating a multi-hazard event set of wind speed, precipitation, and sea level pressure in the Bay of Bengal. The Bay of Bengal is the largest bay in the world and its coastline is home to some 500 million people. It has also been the site of 26 of the 35 deadliest tropical cyclones ever recorded. Compounding factors make the bay especially vulnerable: warm seas provide energy for powerful storms while the shallow, concave bay funnels storm surges up to the coastline. During Cyclone Bhola in 1970, the deadliest storm in history, the storm surge was estimated at 10.4 m high, and half a million people were killed (Biswas, 2020). Tropical cyclones are multi-hazard events; here, we model three of the major drivers of tropical cyclone damages: strong winds, which damage human and natural structures and push waves and storm surge onshore; precipitation, which drives pluvial and fluvial flooding; and low sea-level pressure, which is a major driver of storm surges.

### 4.1 Implementation details

#### 4.1.1 Footprint extraction

We use historical gridded climate data from the ERA5 reanalysis product from 1940 to 2022 (Hersbach et al., 2023). From this, we extract the northerly and easterly 10 m wind speeds ($\mathrm{ms}^{-1}$), total precipitation (m), and sea level pressure (Pa) over the region defined by the bounding box in terms of latitude and longitude. We calculate overall wind speed as the $\ell_2$-norm of the northerly and easterly wind speeds. To aggregate along the temporal dimension, we define a function $h_{k|t}(\mathbf{x})$ for each variable $k = 1, 2, 3$. For wind speeds, $h_{1|t}(\mathbf{x}) = \max_{|t}(\mathbf{x}_1)$, for precipitation $h_{2|t}(\mathbf{x}) = \sum_{|t}(\mathbf{x}_2)$, and for sea level pressure $h_{3|t}(\mathbf{x}) = \min_{|t}(\mathbf{x}_3)$. As a simple deseasonalization function $s_{|t}(\mathbf{x})$, we calculate the monthly medians for each field and subtract these from the data, creating a time series of anomalies. This simplistic deseasonalization method risks neglecting multiplicative seasonality or long-term trends but suffices for the current demonstrative purposes.

We define a risk functional $r_{|ijk}(\mathbf{x})$ as the peak daily 10 m wind speed. This choice prioritizes strong, localized wind storms, such as tropical cyclones, over more widespread, low-intensity storms. Since $r$ is a function of the wind speed only, the extracted precipitation and pressure fields will be conditional on the occurrence of strong winds. If, however, a more nuanced storm definition was required, $r$ could be defined as some function of all three variables.

#### 4.1.2 Probability integral transformations

To fit parametric distributions to the exceedances, we assume a Weibull distribution for wind speeds (see Sect. 3.2) and Generalized Pareto (GPD) distributions for the precipitation and mean sea level pressure. Figures 4a–4c show the fitted parameters for each of the wind, precipitation, and sea-level pressure event sets over the Bay of Bengal. Suitable thresholds above which to extract the exceedances were selected using the modified p-values method of Bader et al. (2018). For wind and precipitation, a threshold was found for all pixels, for sea level pressure, 30 pixels had significant p-values for all thresholds and these were fitted with empirical distributions instead. In the figure, we can see strong spatial coherence across the domain for each climate



variable. Notably, all variables show higher variability near the coast, and wind and precipitation have higher thresholds off-shore. The Weibull shape parameter for wind speeds is in the range of 0.9–1.2 across the domain. This is slightly lower than estimates in other global studies, which have been closer to 2 (Cook, 1982; Harris, 2005). This would suggest that the distribution of wind speeds follows a heavier-tailed distribution for storms in the Bay of Bengal. In some grid cells, the parametric

fit failed repeatedly, and in these locations, only an empirical distribution function was used to transform the data. These are shown as red/white pixels in the fitted parameter plots. For wind speed, precipitation, and sea-level pressure there were one, zero, and 30 failed fits, respectively.

**Figure 4. Left panels** Fitted parameters for the marginal distributions of climate variables over the Bay of Bengal during storms, showing adjusted p-values for Anderson–Darling (AD) goodness-of-fit tests (left column), thresholds $\mu$, scale parameters $\sigma$, shape parameters $\xi$. White pixels indicate locations where the AD test failed and empirical distributions were used instead. **Right column** Density plots of each distribution for a range of tail shapes—with $(\mu, \sigma)$ fixed at $(0,1)$. Shown are: (a) Weibull parameters for peak wind speeds, (b) Generalised Pareto parameters for total precipitation, (c) Generalised Pareto parameters for low pressure.



### 4.1.3 Generative modelling

Using the storm extraction methods described in Sect. 4, we obtained 1249 event footprints from the historical data. Of these,
149 have a maximum wind anomaly exceeding $15\ \mathrm{ms}^{-1}$ and represent storms with very strong spatial coherence. We choose
this wind speed as a threshold above which to define storms as extreme. We initially trained a Wasserstein GAN with gradient
penalty (Arjovsky et al., 2017; Gulrajani et al., 2017) on all 1249 storms; however, we found that the GAN was biased to-
wards the more common, less coherent, storms and basically ignored the extreme storms. Various oversampling regimes were
attempted to address this but training would inevitably collapse once the proportion of resampled extreme storms became too
high.

Instead, we used a modification of the StyleGAN2 with differentiable sample augmentations (StyleGAN2-DA) (Karras
et al., 2020; Zhao et al., 2020). Differentiable augmentation adds semantics-preserving augmentations (e.g. additive noise,
rotations, isometric scaling, saturation changes) to all samples before the discriminator sees them. These augmentations act
as an effective regulariser of the discriminator, forcing it to become invariant to certain distortions and focus on the most
essential data features. The augmentations must be differentiable to allow discriminator gradients to back-propagate all the
way through the generator to the latent space and to guarantee Jensen–Shannon invariance. The StyleGAN2-DA model has
been demonstrated to produce excellent results with as few as 100 training samples, making it a suitable candidate to train on
only the 149 extreme storms.

We trained the StyleGAN2-DA on the 149 extreme storms until it had seen 300,000 footprint images. This took approx-
imately four hours on an nVIDIA GeForce GTX 1080 Ti GPU. We used the trained model to generate 914 multivariate
footprints. Extreme storms in the training data occurred at a rate of $\lambda = 1.82$ storms per year, so this resulted in 500 years of
synthetic hazard events.

### 4.2 Results

Figure 5 shows 16 generated wind footprints for ERA5 training GAN-generated samples. These are presented as they were seen
by the GAN in Gumbel space (5a), in intermediate probability space (5b) and in the actual data space (5c). The corresponding
sea-level pressure and precipitation fields are also shown in the Supplementary Information.

In Section 1 we outlined our criteria for success for the hazard generator. These were that the generated data: (i) replicates
the overall distribution of extreme events, (ii) preserves the dependence structure of the training data across space—in the bulk
and in the tails, (iii) preserves the dependence structure between different variables—in the bulk and in the tails, (iv) preserves
the marginal distributions of the training data, and (v) has a well-justified basis for extrapolation to new, more extreme events.
With (iv) and (v) satisfied by construction, we turn to statistical methods to assess whether criteria (i)–(iv) are satisfied.





**Figure 5.** Comparison of wind speed footprints during storms in the Bay of Bengal for ERA5 training data vs. GAN-generated samples. Shown in (a) Gumbel space, (b) uniform space, and (c) the original data space. The GAN is trained on samples in Gumbel space to emphasise variation in the extremes. The probability integral transform is used to transform the marginals of samples into uniform percentiles, which provide a measure of the extremeness of the wind speed at each point. The fitted parameters from Fig. 4 are used to transform uniform samples back to the scale of the climate variable (anomalies from seasonal median) using the inverse of Eq. 2.1.1

For validation, we only compare the generated dataset with the training set, and omit a validation set. We do this for two reasons: Firstly, given the image generator in a GAN never actually sees the training data, standard overfitting—where the model has excellent performance on training data but performs poorly on new data—is less of a concern than for other machine learning models. This is because if overfitting occurs in GANs, it typically happens in the discriminator. When the discriminator overfits during training its feedback to the generator becomes meaningless. Without useful feedback, the discriminator begins randomly guessing, leading to diverging gradients and training collapse (Karras et al., 2020). Because of this dynamic, an





overfitted GAN usually performs terribly on both the training and validation data and so a validation set becomes less important. Secondly, given we have only $\mathcal{O}(100)$ samples, any validation set would necessarily contain fewer than 50 samples. Such a

small validation set would have high variance, rendering comparison to it relatively meaningless. For these reasons, we consider it most appropriate to only compare descriptive statistics between the training set and the generated samples.

### 4.2.1 Criterion (i): Event distribution

Figure 6 compares the distribution of the generated storms with the ERA5 training data. Wind storms are categorized according to the peak wind speed during each storm. The GAN reproduces the shape of the overall distribution well, but shows some

systemic bias towards producing lower intensity storms, predicting more storms in the 15–20 $\mathrm{ms}^{-1}$ band (3.28% versus 0.00%) and fewer in the 40–45 $\mathrm{ms}^{-1}$ band (0.22% versus 3.36%). On average, the GAN underestimates the occurrence of storms in the 30–45 $\mathrm{ms}^{-1}$ bands by 4.457% and overestimates the occurrence of storms in the 15–30 $\mathrm{ms}^{-1}$ bands by 4.457%. The exact source of this bias remains unclear: it could result from the location or shape parameter of the Weibull parametric fits, however this is unlikely as the Anderson–Darling test returned few significant p-values—indicating good fits—and the low

shape parameter estimates would indicate higher (rather than lower) wind speed estimates (see Fig. 4). Equally, the bias may be a result of the StyleGAN training. in future work, it would be interesting to investigate whether this can be mitigated with longer training times.

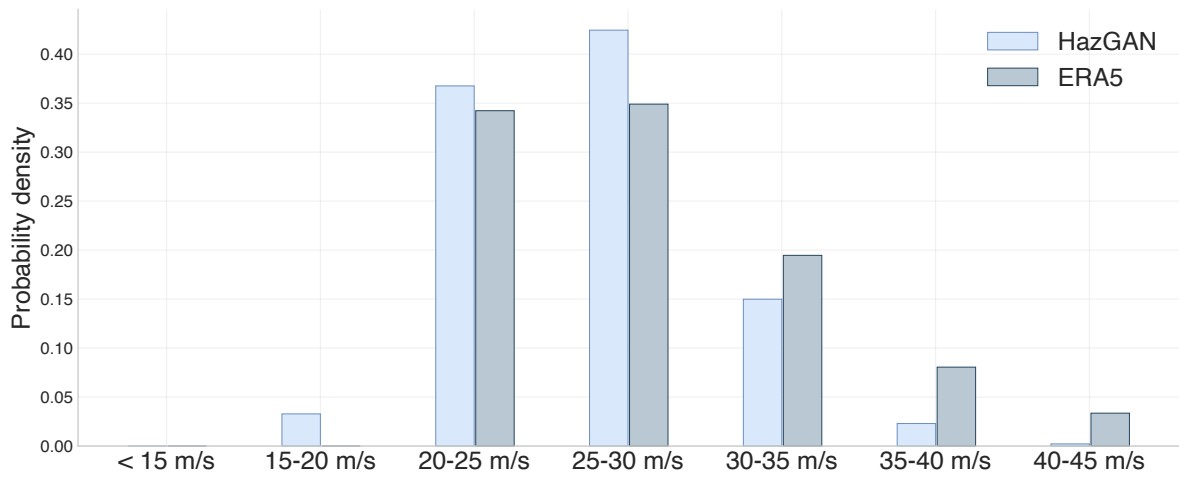

**Figure 6.** Comparison of the distribution of storm intensities in the Bay of Bengal between 500 years of GAN-generated and 81 years of ERA5 storm footprints. Storms are categorized according to the peak wind speed during each storm. While the GAN captures the general shape of the distribution, some systemic underestimation is evident. Implications of this are discussed in the text.

For additional context, we note that ERA5 has well-documented biases when representing tropical cyclones. In a previous study of TCs impacting Bangladesh, ERA5 peak wind speeds were observed to be approximately 20 $\mathrm{ms}^{-1}$ slower than their

IBTrACS counterparts. This is largely accepted to be caused by the coarse horizontal grid spacing (approx. 30 km) of ERA5




simulations, which prohibits the resolution of peak winds near the eye of a cyclone—though Dulac et al. (2024) notes that the sea-level pressure–wind speed relationship appears degraded in ERA5 compared to earlier products, suggestive of further issues with the model physics. Additionally, our model's risk functional extracts events based on their peak wind intensity, and the deseasonalization is based on the climatological median for each month. This doesn't account for the North Indian

Ocean's bimodal tropical cyclone seasonality. A more specialized deseasonalization method or risk functional to detect tropical cyclones or only using training data from the pre-and post-monsoon cyclones, when TC activity is greatest may yield slightly more higher intensity cyclones. However, the fundamental issue is the training data.

These ERA5 limitations highlight a key constraint of our method: it is only as good as the data it is trained on. Any biases or errors in the input data will necessarily be propagated through the whole model. For this reason, it is important to be familiar

with the strengths and limitations of the training data and to consider applying bias correction or downscaling methods to any input data. There may also be better datasets available for specific regions. In the North Indian Ocean, for example, there is the Indian Monsoon Data Assimilation and Analysis (IMDAA) product, which has a higher horizontal resolution of $0.12°$ (12 km). We have also observed it to produce marginally higher winds during Tropical Cyclone Amphan than its ERA5 counterpart. However, since the goal of this work is to develop a flexible methodology applicable to any gridded climate data, we prefer

to use a globally available dataset and consider bias correction to be beyond the scope of this paper. Given our model has successfully reproduced the distribution of different wind storm categories, we consider criteria (i) satisfied.

### 4.2.2   Criterion (ii): Spatial dependence

To check the model is learning the spatial dependence structures, we calculated Pearson correlation and the tail dependence coefficient for wind speed between all pairs of pixels across the domain for the training and generated data (Fig. 7). The plots

for both the Pearson correlation and the extremal correlation are almost identical between the training (a) and generated (b) sets. The same result was observed for precipitation and sea-level pressure fields. We quantify the level of agreement between the ERA5 and GAN-generated correlation structures by calculating the Pearson correlation and mean absolute error between the two correlation fields. For wind, a correlation coefficient of 0.971 (MAE = 0.062) is achieved between the two Pearson correlation structures, showing excellent overall agreement. Results are similar for precipitation (r = 0.981, MAE = 0.054), and

sea-level pressure (r = 0.980, MAE 0.044), with average spatial correlation across three variables of 0.977 (MAE = 0.053). For wind tail dependence coefficient fields, the correlation between the ERA5 and GAN-generated fields is 0.968 (MAE = 0.042). For precipitation tail dependence coefficient fields, the correlation between the ERA5 and GAN-generated fields is 0.906 (MAE = 0.083). For sea-level pressure tail dependence coefficients, the correlation between the ERA5 and GAN-generated fields is 0.948 (MAE = 0.046).

### 4.2.3   Criterion (iii): Multivariate dependence

The final criterion to assess is that of dependence between the different climate fields. To assess this, we again calculate Pearson correlation and the tail dependence coefficient, but between wind and precipitation at each pixel in the domain for the training and generated data (Fig. 8). Again, it is clear that the model captures the dependence structure in the bulk of the data





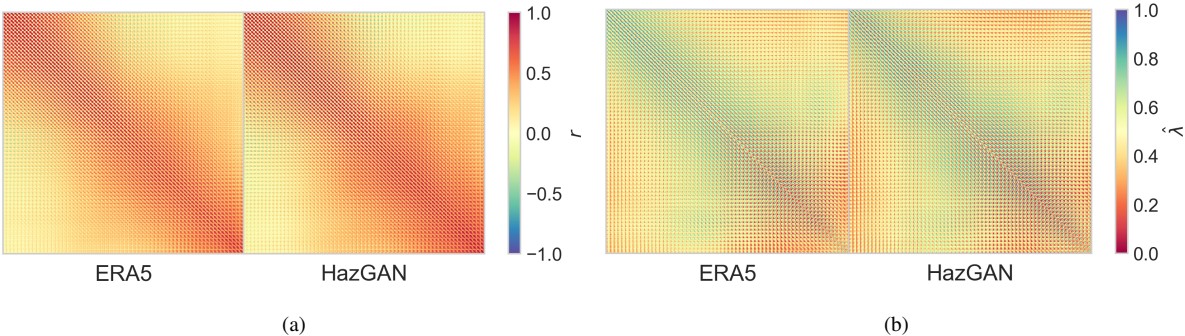

(a)
(b)

**Figure 7.** Pairwise spatial (a) Pearson correlations and (b) tail dependence coefficient estimates for 10 m wind speed anomalies during storms across the Bay of Bengal. For the tail dependence coefficients: values of 1 indicate total dependence and values of 0 total independence in the extremes. The spatial dependence structure of the training data is reproduced by the GAN. In both cases, dependence between pairs of points is dominated by proximity: points along the diagonal exhibit higher correlations.

almost perfectly and captures the general pattern of the extremal dependence structure. The correlation between the ERA5 and
GAN-generated correlation fields for wind and precipitation is 0.918 (MAE = 0.083) and the spatial correlation between the tail dependence fields is 0.692 (MAE = 0.07). For wind speed and sea-level pressure this spatial correlation is 0.647 (MAE = 0.083) between the correlation fields and 0.866 (MAE = 0.099) between the tail dependence fields. For precipitation and sea-level pressure the spatial correlation is 0.887 (MAE = 0.063) between the correlation fields and 0.875 (MAE = 0.120) between the tail dependence fields. Averaged between the three pairs of variables, the average spatial correlation between correlation
fields for variable pairs is 0.817 (MAE = 0.053) and the average spatial correlation between tail dependence fields for variable pairs is 0.817 (MAE = 0.096).

The extremal dependence structure across the domain is noteworthy because high winds and heavy precipitation appear to show more extremal dependence over land than offshore; however, they show more overall dependence approaching the north-east coastline of the Bay. The increased dependence between wind and precipitation near the coast is reasonable; the
strong correlation near the north-east coast corresponds to the prevailing south-westerly winds during the summer monsoon season from June to September. Warm, moisture-laden air blows across the bay from the south-west and, as it interacts with the friction from the land, creates horizontal convergence, leading to air being pushed upwards. The air cools as it rises, leading to precipitation. The stronger the winds, the more horizontal convergence occurs, leading to increased dependence between wind and precipitation. Orographic lifting and land–sea temperature gradients can also occur at these windward slopes and increase
dependence between wind speed and precipitation.

To examine how the model has learned the pairwise relationships between different points in space, Fig. 9 shows scatter plots of the climate variables at two pairs of points: Chittagong and Dhaka—two cities in Bangladesh, and two RAMA (Research Moored Array for African-Asian-Australian Monsoon Analysis and Prediction Atlas) buoys. When comparing the 149-sample





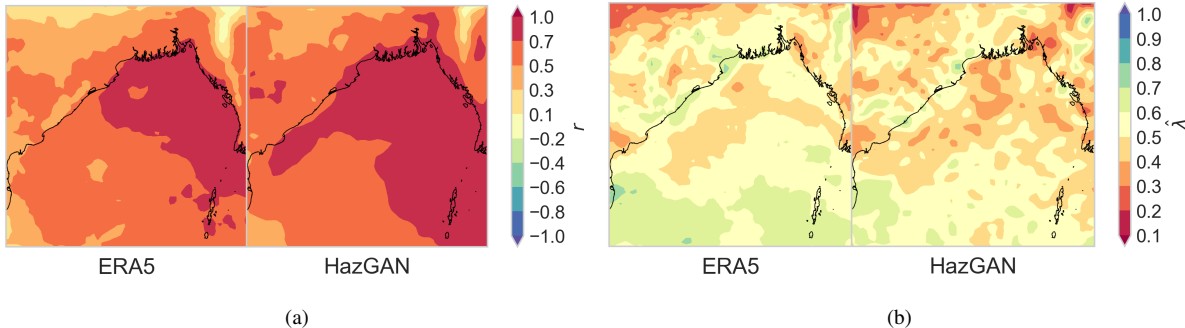

(a)                                        (b)

**Figure 8.** Inter-variable (a) Pearson correlation and (b) tail dependence coefficients for 10 m wind speed and total precipitation over the Bay of Bengal. Similarly to Fig. 7, the GAN reproduces the dependence structure between variables in both the bulk and the extremes. Winds and precipitation show a high Pearson correlation in the north-east portion of the bay while dependence between the extremes of the two variables increases over land. The GAN shows some overestimation of the correlation between wind and precipitation over the bulk of the bay.

training set with 914 generated samples (500 years of samples), it is clear that the model is learning the bivariate density
distribution between each pair of variables, and (in most cases) extrapolating accordingly.





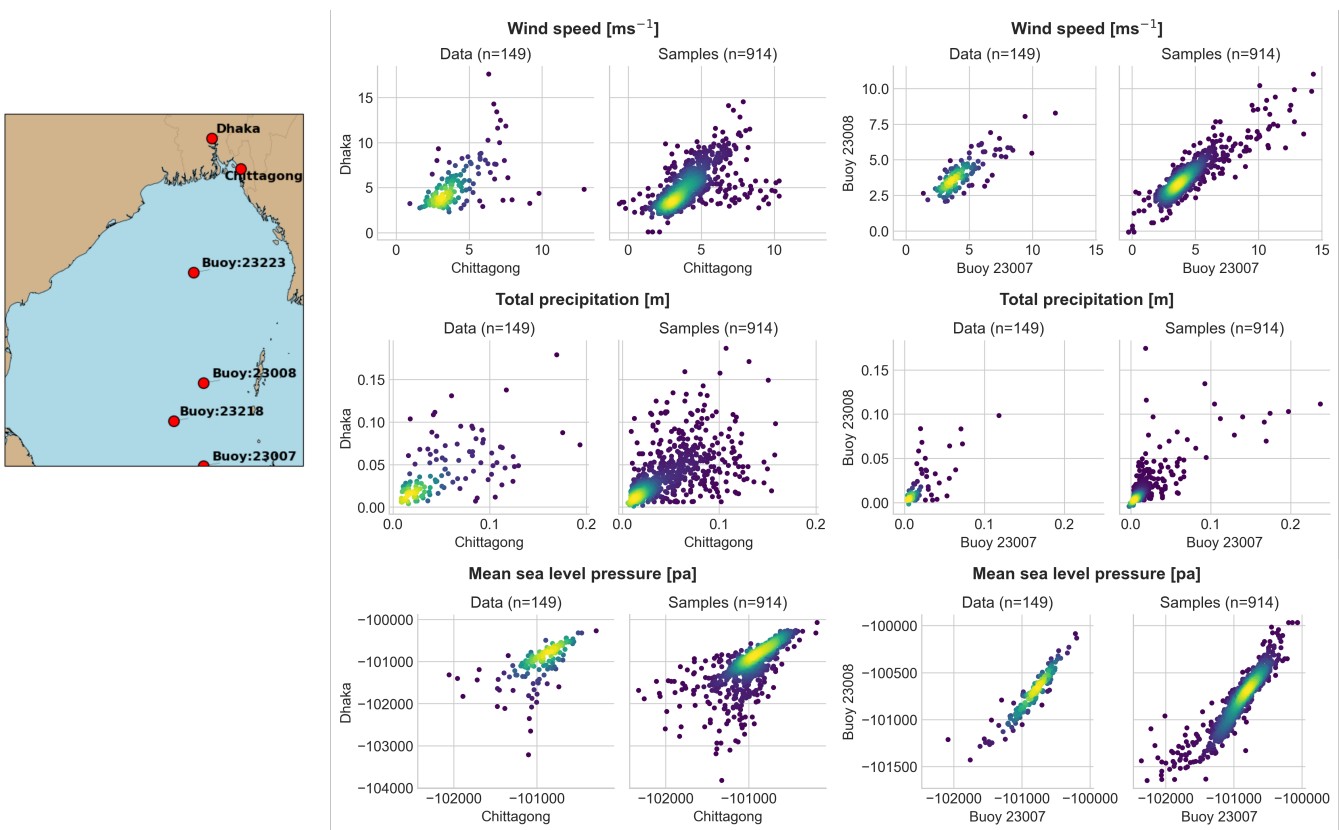

**Figure 9.** Scatter plots comparing the bivariate distribution of monthly anomalies for al three variable between pairs of points during storms. Results are shown for between Chittagong and Dhaka, two cities in Bangladesh, and two ocean buoys. The GAN learns the overall shape of each bivariate distribution, performing particularly well for mean sea-level pressure for both point-pairs.



# 5 Use case: Spatial risk assessment for mangrove forests in the Bay of Bengal

Here, we use our model to estimate risk to mangrove forests from storms hitting the Bay of Bengal. To model storm risk to mangrove forests, we need to model the joint distribution of both wind and precipitation and to estimate aggregate impacts over a domain as large as the Bay of Bengal, we need a model that generates spatial events with the right dependence structure. Thus,
this problem necessitates a new approach to event set generation that can account for spatial and multivariate dependencies, making it a suitable test case for our model. The rest of this section describes the modelling approach and the results.

## 5.1 Implementation details

To estimate damages to the mangroves from the multivariate storm footprints we use a bivariate logistic regression model trained on global historical mangrove damages and tropical cyclone characteristics (Mo et al., 2023; Bunting et al., 2022).
The model predicts the probability that a mangrove patch is damaged, conditional on local winds and precipitation. A patch is defined as "damaged" if it experiences a drop in enhanced vegetation index (EVI) exceeding 20% in the aftermath of a storm. Further details of the method mangrove vulnerability model and relevant calculations are provided in the Supplementary Information. We use the 500 years of wind and precipitation footprints generated in Sect. 4 as input to the mangrove vulnerability model.

To illustrate the implications of ignoring the spatial dependence structure of climate hazards, two more synthetic datasets are constructed: a dataset that ignores all dependence across the region (independence assumption), and a dataset that assumes total dependence across the region (total dependence assumption). The total dependence assumption is the implicit assumption when return period hazard maps are treated as events, while the independence assumption is the assumption implicit when regional risks are modelled separately (Metin et al., 2020).

## 480 5.2 Results

Figure 10 shows the risk profile for widespread mangrove damages over the Bay of Bengal, plotting expected total damage area against return period. Return period is calculated as a function of the total mangrove damages (see the Supplementary Information for calculation details). The figure also displays risk profiles for hazard events generated under independence and total dependence assumptions, which clearly introduce significant bias even at small return periods.

Applied to the ERA5 data, the logistic model predicts 2432.6 $\text{km}^2$ (25%) of the 9917 $\text{km}^2$ mangrove forest in the region to be damaged by a five-year storm event and 2949.80 $\text{km}^2$ (30%) to be damaged by a 100-year storm event. A five-year storm generated by the GAN produces damages of 2335.23 $\text{km}^2$ (24%) and a 100-year storm damages 2799.74 $\text{km}^2$ (28%). A 100-year storm under the total dependence assumption predicts damage to 42% of the mangrove forest in the Bay of Bengal. For a 500-year event, the GAN-generated data predicts damage to 31% of mangrove forest in the region (3026.77 $\text{km}^2$); the
dependence assumption-generated data predicts damage to 47% of mangrove forests (4643.00 $\text{km}^2$); and the independence assumption-generated data predicts damage to only 19% of the mangrove forests (1903.82$\text{km}^2$).





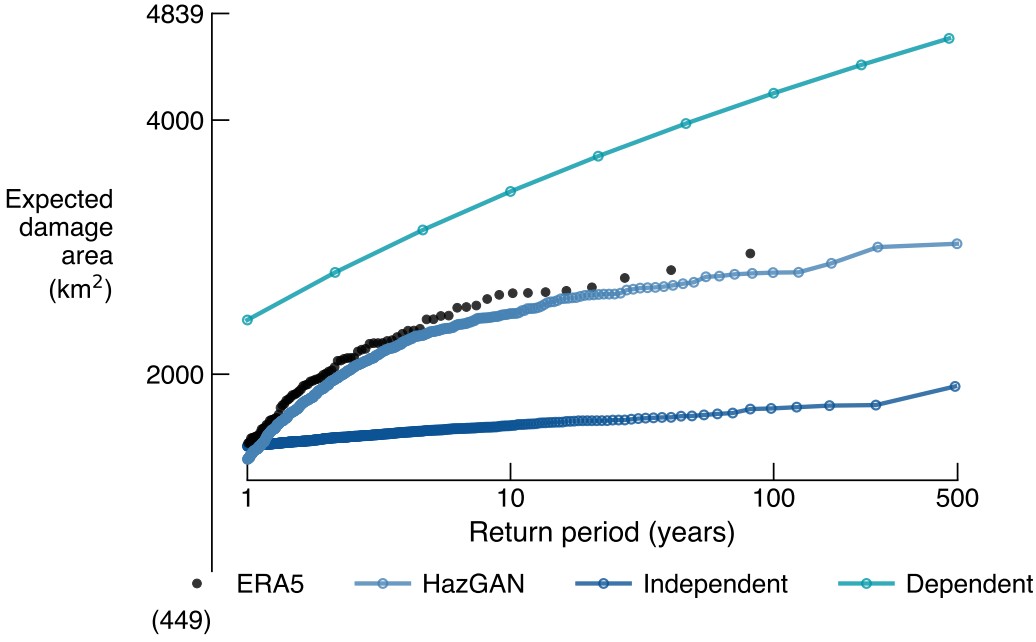

**Figure 10.** Risk profiles showing the expected total area of mangrove forest damaged by storms of different return periods for ERA5 and GAN-generated storm footprints. Also shown are risk profiles for the two simplifying assumptions of total dependence or independence across the domain, where a total dependence assumption is analogous to using return period maps to calculate risk profiles. With a total area of mangrove forest across the bay of 9,917 $km^2$ in 2020 (Bunting et al., 2022), 2,000 $km^2$ corresponds to approximately 20% and 4,000 $km^2$ corresponds to approximately 40% of the total area of mangrove forest present.

Table 1 reveals that while the GAN-generated dataset systematically underestimates total expected damages with a mean absolute error of 93.57 $km^2$ across all return periods (reaching up to 160 $km^2$ for 10-year events), this bias remains an order of magnitude smaller than the independent dataset's systematic underestimation (mean absolute error of 460.45 $km^2$) and

the dependent assumption dataset's systematic overestimation (mean absolute error of 1056.90 $km^2$). These results reinforce findings from Lamb et al. (2010) and Metin et al. (2020), highlighting the critical importance of explicitly modelling spatial dependence of hazards in risk analysis, as treating return period maps as real events leads to massively inflated total damage estimates.

To visualise the qualitative difference between modelling the dependence structure and assuming total or no dependence,

Fig. 11 shows, for the ERA5 and synthetic datasets, a sample corresponding to a 1-in-20 year return period.

Realistic events have clustering in the extremes while the dependent assumption (hazard maps) distributes extreme winds evenly across the entire region and results in far more widespread and unrealistic disruption estimates. The independence assumption shows no spatial coherence and so underestimates the impacts of spatially coherent events.



**Table 1.** Expected damage area and return period deviations for GAN, independent, and dependent generated samples.

| Method | Deviation | 5-year | 10-year | 25-year | 50-year | 100-year | MAE |
|---|---|---|---|---|---|---|---|
| HazGAN | Expected damage area (km$^2$) | -97.37 | -160.35 | -126.58 | -95.87 | -150.06 | 93.57 |
| | (Return period; years) | (-0.07) | (-0.24) | (-2.31) | (8.97) | (17.94) | (0.02) |
| Independent | Expected damage area (km$^2$) | -874.38 | -1042.82 | -1118.72 | -1147.79 | -1219.34 | 460.54 |
| | (Return period; years) | (-0.10) | (-0.21) | (-2.72) | (8.15) | (16.29) | (0.00) |
| Dependent | Expected damage area (km$^2$) | 702.60 | 800.72 | 958.65 | 1153.64 | 1261.63 | 1056.90 |
| | (Return period; years) | (-0.45) | (-0.19) | (-5.64) | (5.64) | (18.46) | (0.64) |

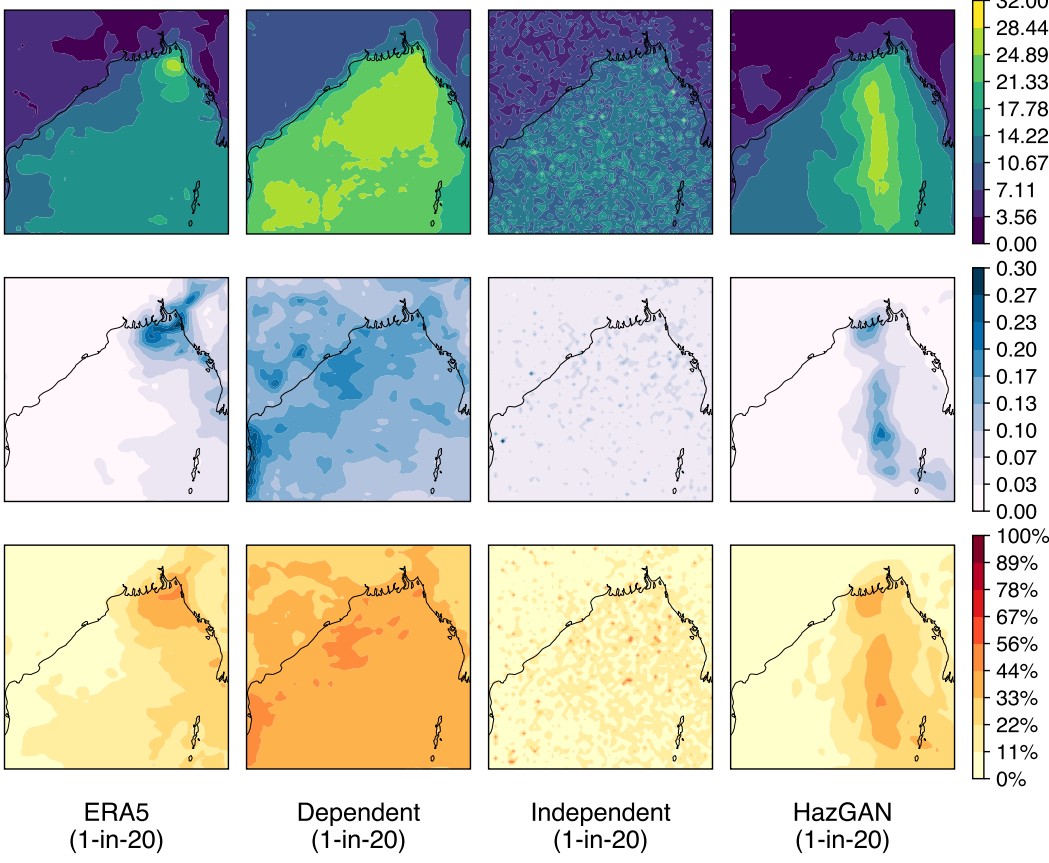

**Figure 11.** Footprints for events with approximately 1-in-20 year return period mangrove damages for the ERA5 and hazGAN-generated datasets, as well as samples from the two simplifying assumptions of total dependence or independence between all variables.



## 6 Discussion

### 6.0.1 Method performance and validation

In this paper, we have demonstrated an efficient method for generating multivariate and spatially coherent event sets, composed of three-variable hazard footprints. The event sets are suitable for risk analysis over large geographic areas. Our three-phase workflow combines statistical extreme value theory with generative deep learning, creating a comprehensive framework for synthetic hazard generation. The computational workload of this method is front loaded during the parameter-fitting and GAN-training steps, which take less than a day. Once trained and fitted, generating new event sets is fast and efficient, generating 5,000 samples in under 30 minutes.

Our validation against three key criteria demonstrates the method's effectiveness. The model successfully reproduces the training distribution of storm intensities and maintains both spatial and multivariate dependence structure across the bulk of the data and in the extremes. The Anderson–Darling goodness-of-fit test validates the parametric fits used to guide extrapolation in the extremes, ensuring a solid statistical foundation for new, more extreme, synthetic events.

### 6.0.2 Practical applications and flexibility

Our method utilises the power of deep learning models in high dimensional settings to push the boundary of the complexity–scope trade-off that exists in risk modelling problems; scientists can use this method to model both multiple variables and large domains, enabling more holistic risk analyses. This is particularly relevant to systems distributed over large areas, where compounding climate hazards represent a significant risk with poorly quantified likelihoods.

The framework's versatility extends well beyond the case study provided: the modular design allows practitioners to substitute wind speed, precipitation, and sea level pressure variables with any climate variables for which training data is available. Hence our method can be used to assess risk from various compound hazards including drought, heat stress, fire risk, and compound coastal flooding, each requiring different combinations of meteorological drivers.

This event-based approach is particularly suited to short-duration, short-memory hazards. Longer duration events, such as droughts, require additional modelling considerations. However, the framework's flexibility accommodates different temporal aggregation periods for variables, so it is possible to include variables accumulated over longer antecedent periods for more complex modelling.

### 6.0.3 Current limitations and data quality

The primary limitation of the method remains training data availability and quality. ERA5, despite being the most comprehensive global reanalysis dataset available, introduces biases through its 0.25° horizontal resolution that smooths extremes. Regional variations in data quality introduce additional uncertainties that propagate through the model. Practitioners should carefully assess their input data's limitations, as any biases will be inherited by the synthetic event set.



### 6.0.4 Extensions

Several promising extensions could expand the framework's capabilities: different deep learning models, dataset topologies, alternative parametric distributions for the marginals, and various data pre-processing methods could all broaden the method's scope and effectiveness.

Alternative deep learning models and architectures could provide better or equal performance to the StyleGAN model used in this paper. Normalising flow models allow for the explicit evaluation of a sample's likelihood, allowing for in-built estimates of training and generated sample return periods. Additionally, different data topologies could be used. Gridded climate data provides a good illustrative example, as it is high dimensional and can be used with image generator models; however, data with network topologies (river networks) or point data (weather stations) could be used with alternative deep generative model types.

Incorporating alternative parametric distributions could build additional flexibility. Circular distributions are an interesting avenue to explore, and could be used to model periodic variables such as wind direction or seasonality. Discrete distributions, such as the discrete generalised Pareto distribution (Hitz et al., 2024) could be used to model extremes in discrete variables such as storm duration.

Data pre-processing could also be used to address biases and uncertainties in the training data. Bias correction and downscaling methods could be used to downscale training data to higher resolutions and correct known biases. The StyleGAN architecture has capacity to train up to 1024×1024 resolution images, suggesting potential for global-scale analysis or higher-resolution simulations. These methods could even be used with coarser global climate model simulations to inform scenario modelling for future risk assessments.

## 7 Conclusion

This work presents a novel framework that successfully combines extreme value theory with generative deep learning to produce multivariate, spatially coherent climate hazard event sets. This method addresses a critical gap in climate risk analysis by generating synthetic events that preserve complex dependencies between variables and across space whilst remaining computationally efficient for practical applications.

The key contribution lies in the three-phase workflow that transforms gridded climate data into multivariate hazard footprints, suitable for input to a deep learning model. The transformation of the data's marginals to a standard Gumbel distribution places emphasis on the extremes of the climate variables during training, ensuring sensitivity to variation in the extremes. This approach combines the statistical rigour of univariate extreme value theory with the high-dimensional capabilities of deep learning methods. Our validation demonstrates that the method can accurately preserve the storm distributions, spatial correlations, and multivariate dependencies essential for accurate risk modelling.

The practical value of this method is evident from its broad applicability across hazard types and its computational efficiency once trained. The Bay of Bengal case study illustrates how practitioners can use this method to generate comprehensive event





portfolios for stress-testing vulnerable systems like mangrove forest ecosystems, but the approach extends readily to other hazards, regions, and asset types.

While current limitations centre on training data quality and availability, ongoing advances in the climate data processing and the framework's inherent flexibility position it well for addressing increasingly complex climate risk challenges. The method
offers climate risk scientists and decision-makers with a powerful tool for generating the comprehensive event sets necessary for holistic climate risk assessments in an era of increasing climate variability.

Future applications will focus on developing more use cases, developing higher-quality training data, incorporating seasonal or directional variables, and extending the framework to other data topologies and deep learning architectures. As climate risks continue to evolve, this framework offers a scalable foundation for developing more sophisticated and comprehensive hazard
modelling capabilities.





*Code and data availability.* The code and data to used will be made available at 10.5281/zenodo.15838238.

*Author contributions.* AP and JH conceptualized the paper and developed the methodology. AP conducted the investigation with supervision from JH and support from YM. YM provided data and code towards the final mangrove damage study. AP prepared the original draft including all code and visualizations. AP, JH, and YM reviewed and edited the manuscript.

*Competing interests.* The authors declare no competing interests.

*Acknowledgements.* This work was funded by the UKRI Engineering and Physical Sciences Research Council (grant number: EP/T517811/1).

The authors would like to thank the Geoff Nicholls, Philip Hess, Shruti Nath, Benjamin Walker, and Alberto Fernandez Perez for their advice at various stages along this project.



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
