# Peer review of "Simulating multivariate hazards with generative deep learning"

_EGUsphere, 2025_

## Author Comment (AC1)

**Benchmarking study**

**The conditional exceedance model**

To provide further benchmarking, we compare our results against the conditional exceedance model of Heffernan and Tawn (2004). The method uses a semiparametric model for the marginals and disentangles the marginal distributions from the dependence structure by transforming all margins to unit Gumbel distributions prior to learning the dependence structure, similar to our approach.

Since we use similar methods to fit the marginal distributions, we only compare our model to the Heffernan and Tawn (2004) model for its ability to model the dependence structure between variables. To do this, we supply the model with the training dataset of storm events, already transformed to a uniform distribution as described in the main text. We modify the *marginalDependence* and *predixt.mex* functions from the R *texmex* package to work with uniform-distributed input and output data. *I.e.*, we remove the generalised Pareto distribution fitting and transformations from the functions. The method models the conditional distribution of a set of variables, conditional that the conditioning variable exceeds some extreme threshold. The objective function is,

$$Q_{|i}(\theta_{|i}, \lambda_{|i}) = -\sum_{j \neq i} \sum_{k=1}^{n} \left[ \log\left(\sigma_{j|i}(y_{i|i,k})\right) + \frac{1}{2} \left( \frac{y_{j|i,k} - \mu_{j|i}(y_{i|i})}{\sigma_{j|i}(y_{i|i})} \right)^2 \right], \tag{S1}$$

with

$$\mu_{j|i}(y) = a_{j|i}(y) + \mu_{j|i} b_{j|i}(y)$$
$$\sigma_{j|i}(y) = \sigma_{j|i} b_{j|i}(y)$$
$$a_{j|i}(y) = a_{j|i} y + I_{\left[a_{j|i}=0, b_{j|i}<0\right]} \cdot \left[ c_{j|i} - d_{j|i} \log(y) \right]$$
$$b_{j|i}(y) = y^{b_{j|i}},$$

where $y$ are unit Gumbel distributed, $i$ is the index of the conditioning variable, $j$ is the index of the dependent variable and $|i$ denotes all the dependent variables. The model is fit for each conditioning variable $y_i$ by optimising $Q$ via the parameters $\lambda_{|i} = (\mu_{|i}, \sigma_{|i})$ and $\theta_{|i} = (a_{|i}, b_{|i}, c_{|i}, d_{|i})$. See Heffernan and Tawn (2004) for a full discussion of the method.

**Benchmarking method**

Since the hazGAN model was trained on a subset of the 150 storms with peak wind speed anomaly exceeding $15 \text{ ms}^{-1}$, we compare results for the dependence fits on this subset only.

To benchmark, we randomly sample 1,000 locations from the training data domain. For each point, we fit a conditional dependence model between wind speed, precipitation, and sea-level pressure, conditioning on the wind speed. From the fitted model, we sample 914 points and calculate the tail dependence coefficient between all pairs of hazard variables. We compare this to the tail dependence coefficients between variables for the training data and the hazGAN-generated data.

We assess the spatial dependence similarly. For each of the 1,000 sampled locations, we randomly sample a second point on the domain with which to compare the spatial dependence fits. We fit a conditional dependence model between each pair of points for each of the three hazard variables, conditioning on the original point, and simulate 914 samples from each of the fitted models. We compare the tail dependence metrics of data simulated by the conditional exceedance model to the training data and to those generated by hazGAN.

Note that in each case with the conditional exceedance model we have fitted a two- or three-variable multivariate model, while the hazGAN model was trained once on all variables simultaneously.

**Benchmarking results**

Figure shows the results for the multi-hazard benchmarking. A scatter plot of the empirical tail dependence coefficient estimates ($\hat{\lambda}$) calculated from training data is plotted against coefficients calculated from 914 synthetic samples, for each of the 1,000 benchmarking locations. For the wind versus precipitation and wind versus sea-level pressure (negated), we see good agreement between all models, with the conditional exceedance model displaying higher variance in the fits. The conditional model shows some bias towards underestimating the tail dependence while hazGAN model shows some bias towards overestimating it. For data points with lower tail dependence between hazard variables ($\hat{\lambda} < 0.6$), the hazGAN model displays some overconfidence, and the spread of points does not cross the perfect fit line (red). Some further investigation may be warranted here, to determine the source of this overestimation of inter-variable dependence.

The left bottom subplot shows the simulated values of total precipitation and sea-level pressure when conditioned on wind speed. In this case, no dependence between the variables was modelled, so the poorer performance is expected. To model the dependence structure between precipitation and sea-level pressure we would need to fit another model, conditional on one of the two variables.

[Figure]

**Figure S1.** Comparison of multi-hazard fits for the conditional exceedance model and hazGAN. Scatterplot shows the multi-hazard tail dependence metrics at 1,000 points in the domain between the training data ($x$-axis) and the synthetic data ($y$-axis). Metrics: Correlation coefficient (r), $R^2$ metric (r2), Spearman correlation coefficient ($\rho$), and mean-absolute-error (MAE).

50  Figure compares the same metrics for the spatial extremal dependence for 1,000 pairs of points in the domain. These are calculated between two different locations for the same variables, conditioning the second point on the first. All models display good fits, with correlation coefficients exceeding 0.9 and mean-absolute-errors below 1%. Again, the hazGAN shows marginally lower variance than the conditional exceedance model.

[Figure]

**Figure S2.** Comparison of spatial fits for the conditional exceedance model and hazGAN. Scatterplot shows the spatial tail dependence metrics between 1,000 pairs of points in the domain between the training data ($x$-axis) and the sampled data ($y$-axis). Metrics: Correlation coefficient (r), $R^2$ metric (r2), Spearman correlation coefficient ($\rho$), and mean-absolute-error (MAE).

55  Overall, this benchmarking provides empirical evidence towards the ability of hazGAN approach to learn the extremal structure between many variables and produce results that are competitive with existing multivariate dependence models.

**References**

Heffernan, J. E. and Tawn, J. A.: A conditional approach for multivariate extreme values (with discussion), Journal of the Royal Statistical Society Series B: Statistical Methodology, 66, 497–546, 2004.

---

## Author Comment (AC2)

**Simulating multivariate hazards with generative deep learning: Final author response to reviewers**

We thank both reviewers for their thoughtful and constructive comments. We have addressed these reviews in our response and in significant modifications to our manuscript, which has improved the rigour and communication of our research.

We have responded to the comments of Reviewer 1 in a separate document. This response addresses the comments of Reviewer 2. Section 1 responds specifically to Reviewer 2's comments while Section 2 provides additional experiments to validate and update some of the modelling choices queried by Reviewer 2.

**1 Response to reviewerer comments**

*The paper "Simulating multivariate hazards with generative deep learning" develops a deep generative framework to create synthetic hazard event sets, aiming to capture multivariate and spatial dependencies among climate hazards. The motivation is sound and the topic is important, but there are some conceptual and methodological aspects that could benefit from further clarification or refinement.*

We thank the reviewer for their recognition of the significance of this study and for their constructive feedback, which we will respond to point-by-point below.

**1.1 Theoretical justification and data transformation process**

*The authors claim that the framework "provides a theoretical justification for extrapolation to new extremes", but this claim is not sufficiently supported. Sections 2 and 3 spend extensive space discussing how extremes can be modeled in statistical and generative frameworks. However, it remains unclear how the data transformation process—first fitting semiparametric or empirical CDFs to obtain uniform marginals, and then transforming them into Gumbel space—actually enhances the model's ability to learn or extrapolate extremes. This procedure also appears inconsistent with prior literature (which they also list). The paper on ParetoGAN demonstrates that a GAN's generated tails inherit the shape of its latent distribution, motivating the use of a Pareto latent space. And, the paper on evtGAN applies a probability integral transform to map marginals to a uniform distribution before training, thereby easing the learning of extremes. Yet, the present study transforms data into a Gumbel distribution after standardizing to uniform, a step that seems unnecessary and contradictory to the cited works. Moreover, the paper does not adequately justify why Gumbel is the preferred target distribution. Lines 180–195 provide background on limiting distributions of extremes, but this discussion is not clearly linked to the choice of transformation.*

Our claim that the framework provided theoretical justification for extrapolation to new extremes was based on the EVT-guided marginal extrapolation and the ability of the model to capture extremal dependence structures in the data. The results submitted with the original manuscript satisfied these criteria; however, as the reviewer points out, the model was excessively conservative in generating new, more extreme events in the transformed space. This limited our ability to fully demonstrate the advantages of our framework. In subsequent experiments (Sect. 2), we have improved the model's ability to generate more extreme events while maintaining these properties. We will discuss this in more detail in relation to Comment 1.4.

Regarding the transform to Gumbel margins for training, we thank the reviewer for highlighting this important point and agree that we did not sufficiently explain our motivation in the submission. We include a more extensive justification here, which we will incorporate into the revised 'Theory and Methods' section of the manuscript.

Additionally, we have further investigated the effects of different training distributions, with results supporting our decision to deviate from the approaches of Huster et al. (2021) and Boulaguiem et al. (2022), as noted by the reviewer. The experiment results are included in Sect. 2.

An initial point is that standardising margins to Gumbel, Laplace, or unit Pareto distributions is common practice in the literature of multivariate extremes (Heffernan and Tawn, 2004; Healy et al., 2025; Huser and Wadsworth, 2022; Keef et al.,

2009). Copulas are invariant under increasing and continuous transforms and working in different spaces emphasises different properties of the margins and their dependencies (Embrechts et al., 2002). Gumbel or Laplace margins, for instance, linearise the tail dependence structure between variables, which can make subsequent dependence modelling more tractable (Heffernan and Tawn, 2004; Keef et al., 2009; Quinn et al., 2019).

Regarding our work's deviation from other authors, we first address deviations from Huster et al. (2021), who were motivated by the theoretical results of Wiese et al. (2019) that GANs cannot generate heavy-tailed samples using light-tailed latents.[1] Their solution (ParetoGAN) modified GANs to use heavy-tailed latents, requiring a new (non-adversarial) loss function and power transformations to the generator's output. Incorporating this loss function into StyleGAN would be challenging; however, we obtained comparable results by simply transforming the training data to have light-tailed margins (tail index $\xi = 0$, *i.e.*, Gaussian, Gumbel, or Laplace). Aligning with the requirements of Huster et al., this approach too preserves agreement between the asymptotic tail-heaviness of the training and latent variables, while the preservation of the copula dependence structure is guaranteed under continuous and increasing transforms (Embrechts et al., 2002). We provide additional experimental results justifying this simpler approach in Sect. 2.1.

As the reviewer notes, Boulaguiem et al. (2022, evtGAN) transformed margins to a uniform distribution; however, they used a Gaussian latent space with these uniform margins. The theoretical results of Huster et al. (2021) and Wiese et al. (2019) imply that this could create a mismatch between the tail shapes of the latent and training variables. More practically, uniform training margins risk severely under-representing tail variability. On a uniform scale, 0.9 versus 0.99 corresponds to approximately 10- versus 100-year return levels but 0.5 versus 0.51 corresponds to approximately 2- versus 2.4-year return levels. Clearly, equal intervals in uniform space correspond to significant changes in physical space. In Sect. 2.2 we will demonstrate the unrealistically homogenous footprints obtained by training with uniform margins.

The results in Sect. 2 suggest it suffices to transform margins to a light-tailed (tail index $\xi = 0$) distribution for training with Gaussian latents. The choice between Gaussian, Gumbel, or Laplace training margins is less clear-cut and we have not systematically investigated it in these experiments. We hypothesise that extrapolation may depend on the sub-asymptotic behaviour of the data: these distributions will appear different in the sub-asymptotic regime, which may affect what the GAN learns. Gumbel data has the fastest sub-asymptotic tail decay rate ($\sim \exp(-\exp(x^2))$), while Laplace has the most linear decay ($\sim \exp(-x)$) and Gaussian lies in the middle with decay $\sim \frac{1}{2}x^2$. In Sect. 2.2, we observed that the Gaussian-based training performed best (*i.e.*, transforming to uniform pseudo-observations then to Gaussian margins via the probability integral transform). Thus, we will switch to Gaussian margins as the new, final choice of training distribution.

**1.2 Event selection, thresholds, and conditional extremes**

*Besides, the selection of the hazard event (or footprint) is not clear enough. Several thresholds are mentioned but not clearly specified. For example, in L280, "risk functional exceeds some threshold $v_r$" is introduced, which is later set to the peak daily 10 m wind speed exceeding 15 m/s in section 4.1.1. It appears that only wind speed is treated as the peak-over-threshold variable, while precipitation and sea-level pressure (SLP) may not be extreme during these events. If precipitation and SLP are conditionally sampled during extreme wind events, their marginal distributions do not represent natural extremes. It is therefore questionable to apply tail fitting (e.g., GPD in L340–341) or Gumbel transformation to them. The rationale for transforming these non-extreme variables should be clarified. Threshold selection is also vague: phrases such as 'suitable threshold' (L297, L343) are used without explicit criteria. A brief explanation of the Bader et al. (2018) modified p-values method would help readers understand how thresholds were chosen within the transformation process. If these 'suitable thresholds' are required for building GPDs, their purpose and determination need to be stated clearly. Furthermore, if the precipitation and SLP data are conditional on extreme wind events, the validity of fitting GPDs to these conditional subsets is doubtful.*
* * *
[1] The results of Wiese et al. (2019) and Huster et al. (2021) relate specifically to the asymptotic regime. The results were also only derived for piecewise linear networks, so may not be as strict for networks fully nonlinear layers. To the best of the authors' knowledge, this has not been investigated to date.

**1.2.1 Threshold selection**

We thank the reviewer for highlighting the need to clarify the difference between the three threshold methods in the manuscript. We will significantly revise and clarify how these are presented. To summarise, the three different thresholds are:

**Threshold (i):** The threshold for event selection ('some threshold $v_r$', L280) uses runs-declustering (Coles et al., 2001) to identify independent storm events. We use a grid search over threshold and run-length combinations, selecting parameters that maximise the number of independent storms while satisfying a Ljung–Box test for independence of storm maxima. We will include a brief description of Coles et al. (2001)'s procedure in the revised manuscript.

**Threshold (ii):** The 15 m s$^{-1}$ threshold (Sect. 4.1.3) further subsets existing events to focus the GAN on extreme cases. This manages a key trade-off: training on all 1249 events led the GAN to neglect rare extremes, while training on only the most extreme events reduced sample size. We tested alternatives including oversampling rare events (Bhatia et al., 2018), but this distorted the intensity distribution without improving results. The 15 m s$^{-1}$ threshold was chosen by visual inspection: storms exceeding this value consistently showed coherent spatial structures.

**Threshold (iii):** The 4096×3 marginal thresholds above which to fit POT models were chosen automatically using the Bader et al. (2018) method, of which we will add a brief description to the main text as recommended by the reviewer. Below these thresholds, empirical distributions are used to obtain the uniform pseudo observations. This approach is identical to the semiparametric approach of Heffernan and Tawn (2004, Eq. (1.3)).

All three thresholds are used in the data preparation process. Threshold (i) ensures independence of the extracted events, thereby justifying the use of GPD fits to marginal exceedances above the thresholds defined by Threshold (iii).

**1.2.2 Conditional sampling of precipitation and sea-level pressure**

For modelling multi-hazard risk *during windstorms*, conditional sampling of precipitation and sea-level pressure is desired; we are not attempting to model the natural extremes of each variable independently. Rather, we seek to model the joint behaviour of all three variables conditional on a windstorm occurring. If we instead selected events where *any* variable was extreme, we would capture a broader set of hazard events, but would no longer be modelling windstorms specifically. For our application (wind hazard with compound precipitation and pressure effects), conditioning on wind extremes only is appropriate.

Regarding statistical validity of conditional POT fitting, the necessary and sufficient conditions for the Pickands–Balkema–de Haan theorem require that the parent distribution be i.i.d. and lie in the maximum domain of attraction of an extreme value distribution. Conditioning on wind extremes does not violate these requirements, provided the conditional samples remain independent and identically distributed.

Our procedure ensures these conditions through (i) runs declustering to enforce independence between events, and (ii) de-seasonalisation to remove non-stationarity. Therefore, conditional exceedances of precipitation and sea-level pressure during independent windstorm events can validly be modelled using GPDs.

**1.3 Spatial biases in event selection**

*Also, the spatial extent of the risk functional is not clear. It is not specified which grid cells are used when identifying events. If any pixel exceeding 15 m s$^{-1}$ triggers a storm event, some locations will appear in the dataset more frequently and with stronger extremes, while others will rarely be included or will contain only moderate values. This non-uniform sampling may distort spatial dependence structures and could make the transformation inappropriate at locations with few or no extremes. The authors should clarify how spatial selection biases are handled.*

As clarified in Comment 1.2.1, storm events were identified using the runs declustering method of Coles et al. (2001) applied to a spatially aggregated wind metric and the 15 m s$^{-1}$ threshold was used later to further subset that event set. However,

the reviewer raises an important point that this spatially-aggregated metric may introduce bias if some locations trigger storm
events significantly more often than others.

To check whether our selection method introduced spatial biases, we investigated whether certain locations triggered storms more often than others. Figure 1 maps the spatial distribution of mean and variance in wind speeds (left, centre) and the frequency with which each grid cell triggered event selection (*i.e.*, contained the maximum value on a storm day; right). While mean winds and wind variance exhibit strong spatial patterns, the distribution of storm-triggering pixels is relatively uniform across the domain, although predominantly offshore. This uniformity is likely due to the prior deseasonalisation step which subtracted the climatological median from each pixel. Despite higher baseline variability in some regions, event selection appears to capture spatially diverse storm patterns rather than being dominated by a few high-variance locations. Over the 1941–2022 period, 1249 storms were identified, with trigger pixels distributed across the entire domain and no single pixel accounting for more than 16 events (1.2%). Factoring in all days within each storm, this number only rises to 46 events (3.7%). We will include this analysis and Fig. 1 in the revised manuscript to clarify the spatial characteristics of our event selection procedure.

[Figure]

**Figure 1.** The mean 10 m wind speed (left), standard deviation of 10 m wind speed (centre), and event-triggering frequency (right) of each grid cell. The event trigger frequency reflects the number of times a storm was extracted, based on the value of that grid cell.

Overall, however, we believe that some level of spatial bias reflecting the real-world distribution of storm events is acceptable. This is because we are not attempting to model the unconditional joint distribution of wind speeds across all grid cells. Rather, we explicitly seek to model the conditional distribution of spatial hazard patterns given that a storm event has occurred, as per the criteria of our user-defined risk function. Locations regularly exposed to strong winds will contribute to the dataset more frequently and have more extreme distributions, but this correctly reflects the spatial distribution of wind hazard across the region. Conversely, locations that are rarely exposed to extreme winds will have distributions concentrated at lower values— again, this correctly represents their conditional behaviour during major storms. We will clarify this nuance in the revised manuscript.

Additionally, for locations with few or no extremes, the POT threshold selection method reverts to empirical fits where no good GPD fit is obtained for any threshold. In these cases, empirical distribution and quantile functions are used with no extrapolation. This is appropriate because these pixels experienced few or no extremes during the identified storm events—we have no empirical evidence to support generating extreme values at these locations during our defined events.

**1.4 Systematic underestimation of hazard events**

*Regarding the results in both applications, there seems to be a systematic underestimation of the hazard events (as the authors also acknowledge). In Section 4.2.1, the authors rule out some unlikely reasons, stating it might be due to inadequate training. However, the model was trained for only four hours, and longer training could easily have been tested. Inadequate training time is not a convincing explanation for systematic bias, particularly given*

*that the dataset contains only 149 samples. A more likely reason is that the generator fails to capture the true tail behavior, suggesting that the framework does not enhance extremal learning as expected.*

The enhancement in extremal learning claimed in the original manuscript combined three key objectives: (i) that marginal extrapolations in physical space were justified theoretically, (ii) that the correct overall distribution of extreme events was generated, and (iii) that the extremal dependence structures were captured correctly. Objective (i) was satisfied by our approach: the marginal values in physical space were controlled by well-developed extreme value theory. Objective (iii) also appeared satisfied: the spatial structure of the generated extremes showed reasonable agreement with the training data for the metrics tested.

For Objective (ii), we conducted additional investigations and identified that the under-extrapolation was stemming from how the data was pre-processed and rescaled. We found that the final pre-processing step to rescale the data to (0,1) (boundaries excluded) for StyleGAN was not providing the GAN with sufficient headroom for extrapolation. We switched to a new (simplistic) rescaling method—min-max scaling to [0,1] then multiplying by a factor of 0.9. With this rescaling, the model was better able to extrapolate to new extreme values. Extrapolation performance is discussed in more detail in Sect. 2.2 and Fig. 4 visualises the extrapolation in the StyleGAN's [0,1) training space for each training distribution. For Gaussian, Gumbel, and uniform training margins, the model is now producing extrapolated extremes (Fig. 5).

**1.5 Structure and writing**

*Overall, the paper is detailed, with extensive explanation and context, but the information flow is uneven. Some sections (e.g., statistical background) are lengthy yet do not directly strengthen the main argument, while others (e.g., motivation for the threshold selection) are too brief to follow. A clearer organization would improve readability. Although Sections 2 and 3 are labeled "Theory" and "Method", the boundary between them is unclear. The "Theory" section includes methodological descriptions (e.g., semiparametric fitting, Gumbel transformation), while the "Method" section continues with implementation details. Given the close interdependence between the conceptual justification and the procedural steps, it would be clearer and more readable to merge these into a single "Methods" section. This would also allow the authors to explicitly link each design choice—such as the Gumbel transformation, declustering, and marginal fitting—to the overarching goal of modeling extremes, rather than scattering them across two sections.*

We thank the reviewer for this constructive feedback regarding the balance of information. We will expand important discussions regarding motivation for key steps and condense lengthy sections on statistical theory. We will also merge the 'Theory' and 'Methods' section into a single 'Theory and Methods' section and directly link each design choice with the motivating theory.

**1.6 Minor comments**

*L35: The explanation of "total dependence" vs "total independence" could be clarified. A 1-in-50-year map treats each pixel as independent when computing return levels, while interpreting the map as a single event implies total dependence. Clarifying this distinction would avoid confusion.*

We thank the reviewer for pointing this out. "[I]nterpreting the [return level] map as a single event' was indeed our intended meaning for total dependence and we will clarify this in the manuscript.

*L177: The discussion of the bias–variance trade-off in traditional EVT models is interesting. However, deep learning models face similar challenges, and it would be useful to explain how the proposed approach mitigates them.*

Our deep learning approach does not actually attempt to mitigate this limitation of univariate EVT models. This description was rather intended to motivate the use of alternative parametric models for extremes in certain cases. For example in some cases sub-asymptotic models are advocated for modelling extreme winds, *e.g.*, Cook and Harris (2004).

195 *L185-190: The background on limiting distributions is informative, but the connection to the later choice of the Gumbel transformation could be made more explicit.*

Similar to the previous comment, this was also intended as a discussion of the appropriateness of using a GPD to describe wind speeds, rather than relating to the final Gumbel transform.

We will remove L177 and L185-190 they are not central to the main argument of the manuscript. In general, we are confident
the theory–motivation linkings will be made clearer when we follow the reviewer's suggestion to merge the 'Theory' and 'Methods' section and directly link each design choice to its motivating theory.

*L181: Please provide references for the statement "a wealth of evidence".*

This statement refers to Harris (2005), who provide multiple examples of a Weibull parent for wind speeds. Additional references include Harris and Cook (2014) and Cook (2023). L181 will also be removed, however, in line with the previous comment.

*L204: Equation (1) could be explained more clearly. How is it derived? How to interpret physically?*

Eq. (1) is used to allow for non-GPD parametric distributions to be used in the tail regime, for example, the penultimate model of Harris (2009). The sub-threshold term is the standard ECDF, while the tail term subtracts the product of two survival functions—the empirical probability of a threshold exceedance and the parametric probability of the exceedance value—from 1 to generate a semi-parametric CDF. This is a simple generalisation of Eq. (1.3) of Heffernan and Tawn (2004).

We will keep this section but highlight that it is intended for unusual cases where another parametric tail distribution is required.

*L209: F_P appears without prior definition. Is this F_D?*

Yes, we thank the reviewer for catching this typo.

*L225: y_n1, y_n2 could be defined when first introduced.*

We will define $y_1, y_2$ respectively as observations of the Fréchet random variables $Y_1$ and $Y_2$.

*L235: Why 90th percentile?*

The 90th percentile was just intended to serve as a concrete example, we will make this clearer.

*L280: Adding one or two sentences on the declustering algorithm to indicate how it helps to identify hazard events would be better.*

We will add a brief description of the Coles et al. (2001) declustering algorithm.

*L294: Eq.(2.1.1) appears multiple times but cannot be found.*

We thank the reviewer for highlighting this, this should indeed say Eq. (1).

*L237: ERA5 reanalysis monthly or hourly data?*

We will clarify that this is hourly data, which we subsequently resample to daily extremes.

*L340: Typo – exceedances*

We will replace all instances of 'exceedences' with 'exceedances' in the manuscript.

*L357-360: The whole transformation in this framework was aimed at extreme events. If GANs trained on the transformed data were still biased towards more common storms, does it mean the transformation does not function as expected?*

We hope previous responses to the main comments are satisfactory on this front. To summarise: while the original transformation could guarantee theoretical justification for the physical values of new extremes—backed up by EVT—and their dependencies, it could not force the GAN to generate more extreme events in the transformed space. This turned out to be more related to other factors in data preprocessing, which we have now resolved (Sect. 2.2).

*L361-365: Linked to prior comment, does this mean the differentiable augmentation is the technique that solves the challenge of extreme training?*

We will include a more thorough review of how different design choices contributed to solving the challenge in the Discussion of the revised manuscript.

The contribution of DA allowed us to train on a smaller number of samples, which reduced the model's tendency to simply ignore rare samples in favour of the bulk of the data. This improved representation of more extreme events within the generated data, but did not in isolation solve the challenge of generating realistic multivariate spatial extremes. In particular, the model did not start extrapolating until we fixed the data rescaling (Comment 1.4 and Sect. 2.2) and the EVT transforms on the marginals were required to guide transformation back to physical space in a theoretically justified manner. In this way, the GAN takes care of the copula structure, while EVT controls the marginal behaviour. Without the EVT, there is little theoretical justification for the physical values of new marginal extremes generated by the model.

*L369: Does this mean the model was trained for 300,000/149 $\cong$ 2013 epochs?*

Yes, that is correct.

*L374-375: This is not clear. Are these images the input (before processing) to xx space or after processing in xx space?*

We will expand the caption and include sub-captions to make the purpose of each image clearer. Fig. 5 (a) shows the wind footprint samples on which the StyleGAN model was trained in the Gumbel-transformed space,[2] (b) shows these samples transformed back to uniform probabilities or 'pseudo observations' as an intermediate step, and (c) shows the samples transformed back to physical space, *e.g.*, wind speed in m s$^{-1}$ or pressure in Pa.

*L410-412: This sentence could be simplified for readability; briefly define what is meant by a "more specialized" deseasonalization or risk functional.*

We will elucidate what we mean by each of these terms. For a more specialised deseasonalization function one could correct for multiplicative as well as additive seasonality, fit an auto-regressive model to the data, or stratify the data by season. More specialised risk functionals could involve functions of multiple hazard variables or more complex functions over space, allowing us to capture storms according to more specific criteria. There is much flexibility in how these could be defined, but we will highlight some examples to make the core idea clearer.

*L423: What threshold is used to compute the tail dependence coefficient?*

This is calculated over a range of thresholds from 0.75 to 0.99 and the average across these thresholds is reported.

*L475: It is not clear enough how the independence assumption is handled. How are "regional risks" modelled separately?*

The independence assumption refers to cases where risk in nearby regions is modelled without considering dependence between those regions. For example Lamb et al. (2010, Fig. 7) demonstrated the bias in total risk obtained when modelling fluvial flood risk in Leeds and York separately compared to via explicitly modelling their dependence structure. While the complete-independence-between-pixels approach is an extreme example of this, similar approaches actually exist in the literature, *e.g.*, Huang et al. (2025, Sect. 3.3).
* * *
[2]Note the samples are shown before additional rescaling to [0,1].

*Overall, this research is a valuable addition to the scientific literature on climate extremes and compound hazards. In conclusion, I recommend this manuscript for publication after the authors address the points mentioned above.*

We thank the reviewer for this positive feedback and for their overall constructive and thorough commentary. We are confident that this feedback will significantly improve the quality of this work.

**2 Additional experiments**

The following additional experiments will be incoporated into the main manuscript, along with the Brown–Resnick benchmarking experiment conducted in response to Reviewer 1's comments.[3]

**2.1 Comparison to Pareto GAN**

Huster et al. (2021) investigated the ability of GANs to capture the extreme values of heavy-tailed data. In their experiments, they compared a classic GAN with Gaussian latents with their solution: ParetoGAN. ParetoGAN used heavy-tailed latents, a modified (non-adversarial) loss function, and applied power transforms to the generator's output. The details of the original experiments are provided in full in (Huster et al., 2021).

Here we validate an alternative approach: rather than modifying the GAN to have heavy-tailed latents, we transform the training margins to a light-tailed (Gaussian) distribution, preserving agreement between the data and latent variable tails.

Table 1 shows the results for several experiments. The margins field specifies whether the original heavy-tailed (Cauchy to match Huster et al. (2021)) or Gaussian-transformed data was used. The Kolmogorov–Smirnov test statistic is defined as the largest magnitude difference between the distribution functions of two distributions and gives an indication of how well the *modes* of the data match. The area between of the log-log plots of the empirical survival functions of real and generated samples gives an indication of how well the generated *tails* match the real samples.

**Table 1.** Additional experimental results based on Huster et al. (2021)

| Margins | Latents | Seed | KS statistic (↓) | Log-log area (↓) |
|---------|---------|------|------------------|------------------|
| Cauchy | Gaussian | 1000 | 0.0141 | 54.6642 |
| | | 1001 | 0.0185 | 54.5919 |
| | | 1002 | 0.0133 | 52.9095 |
| | | Mean | 0.0153 | 54.0552 |
| Cauchy | Pareto | 1000 | 0.0043 | 5.5712 |
| | | 1001 | 0.0060 | 2.4611 |
| | | 1002 | 0.0103 | 36.6845 |
| | | Mean | **0.0069** | 14.9056 |
| Gaussian | Gaussian | 1000 | 0.0115 | 18.0198 |
| | | 1001 | 0.0132 | 3.8996 |
| | | 1002 | 0.0107 | 5.0781 |
| | | Mean | 0.0118 | **8.9992** |

Figure 2 shows the tail distributions of the training (orange) and generated (blue) samples. Clearly, training heavy-tailed data with a Gaussian GAN under-represents the tails, but aligning the data and marginal tail weights leads to much-improved tail representation, whether it is achieved via ParetoGAN or transforming the heavy-tailed Cauchy training data.[4]
* * *
[3]https://doi.org/10.5194/egusphere-2025-3217-AC1

[4]The code to reproduce these results is available at https://github.com/alisonpeard/paretogan.

[Figure]

**Figure 2.** Tail distributions of real and GAN-generated samples. (Based on Huser and Wadsworth (2022).)

**2.2 Performance using different training margins**

The following experiments demonstrate the effect of training StyleGAN2-DA on data transformed to have different marginal distributions. Holding all other settings constant, we compare the results obtained using Gumbel, Gaussian, and uniform margins for training. For comparison, we also include results from training on min-max scaled data, *i.e.*, standard DCGAN training.

After transforming the uniform pseudo observations to the training distribution (Gumbel, Gaussian, or uniform), additional rescaling was required as StyleGAN works with images bounded in [0,1] space. Previously, this was achieved via a function of the data minima, maxima, and number of samples, but this was sensitive to the dataset size and did not sufficiently shrink the maximum value to provide headroom for extrapolation. Instead, we use a simpler approach and apply min-max scaling to the training data and multiply the result by 0.9 to generate data in [0,0.9].

Figure 3 shows wind footprints generated from models trained using each configuration. Qualitatively, the Gumbel- and Gaussian-based and min-max-scaled footprints best-resemble the ERA5 training footprints. The footprints obtained from a StyleGAN2-DA trained on uniform margins have large patches of very high values and look unrealistic.

**2.2.1 Tail behaviour**

[Figure]

**Figure 3.** Samples generated by training StyleGAN-DA in different marginal space.

**2.2.2 Extrapolation**

To explore how the chosen training distribution impacts extrapolation, Fig. 4 shows the tail regime for the training samples in each training domain, after rescaling to [0, 0.9], providing insight into the nature of model extrapolation without additional interactions with the inverse probability integral transform.

The first observation is that the distribution of the sub-asymptotic training tails directly effects the shape of the generated tails and training data which decays smoothly to zero within [0, 0.9] leads to the cleanest matching of the training data tail shape. For both Gumbel-distributed and min-max scaled data, there is a sharp cut-off in the density at 0.9. The model appears to learn a heavy tail and attempt to extrapolate beyond the headroom it has been provided, leading to a build-up of values at 1.0. We consider the Gaussian results satisfactory for the current scope of work, but future work could explore the effect of more headroom on Gumbel or min-max scaled training data. Interestingly, the generated uniform data shows a smoother density decay to zero; however, the generated values neither align well with the training data nor appear uniform. The similarity of the min-max-scaled and Gumbel results could be down to the fact that the limiting distribution for many climate variables is indeed the Gumbel distribution. Since we are working in the tail region, the min-max scaled data may be already partially converged to its asymptotic Gumbel form, hence the similarity between tail densities.

Overall, the Gaussian-distributed data looks the most reasonable. We hypothesise that this good performance is due to the perfect distributional alignment between the Gaussian latent distribution and the Gaussian training margins.

Figure 5 compares the overall distribution of storm intensities for the real and synthetic storm datasets. Again, there is reasonable agreement for storms generated using Gaussian-, Gumbel-, or min-max-transformed training data. Note that the scaling-based data does not extrapolate beyond the training data. This is because the empirical transforms do not provision for extrapolation. The uniform-based data exhibits significant oversampling of very high values. Here, the Gaussian and Gumbel-based results show the best performance with respect to extrapolation and distributional adherance to the training data.

[Figure]

(a) Min-max scaled    (b) Uniform    (c) Gumbel    (d) Gaussian

**Figure 4.** Headroom generated by training StyleGAN-DA in different marginal space.

[Figure]

(a) Gaussian    (b) Gumbel

(c) Uniform    (d) Min-max scaled

**Figure 5.** Samples generated by training StyleGAN-DA in different marginal space.

**2.2.3 Dependence structure**

This section examines the ability of each model to capture multivariate dependence structures.

Figure 6 compares the bivariate distribution of wind speed at two ocean buoys between the training (left-hand subplots) and generated (right-hand subplots) event sets. The Gaussian and Gumbel-based results show overall good agreement, however there is more variability in the Gaussian-based plot. Again, the uniform-based model performs poorly, showing no extrapolation for this example. The scaling-based data has not extrapolated beyond the training data, as expected, however it does show reasonable agreement in the bulk of the data.

In the main manuscript, we estimated the upper tail dependence using the empirical estimator of Reiss and Thomas (2007, Eq. 2.62):

$$\hat{\chi}(u) = \frac{1}{n(1-u)} \sum_{i=1}^{n} \mathbf{1}\{U_i > u, V_i > u\}, \tag{1}$$

where $(U_i, V_i)$ are pseudo-observations, uniformly distributed on $[0,1]$. To reduce noise, we averaged over thresholds $u \in [0.75, 0.90]$.

[Figure]

**Figure 6.** Samples generated by training StyleGAN-DA in different marginal space for a pair of points buoys in the Bay of Bengal.

We also show the Pearson correlation coefficients and tail dependence coefficients between pairs of hazard variables across the spatial domain.

To calculate tail-dependence metrics for the min-max-scaling-only experiment, we used the empirical distribution function to obtain pseudo-observations, so as to avoid using any extreme value theory. For fair comparison between experiments, we then calculated the pseudo-observations empirically for the other experiments here. We report these instead of the results based on the semiparametric fits from the main text. The results for each of the generated datasets represents the average across five 149-sample subsets of the 914 generated samples. This sub-sampling is carried out in order to avoid differences stemming from the rank metric's sensitivity to sample size.

To view the spatial distribution of these metrics, Fig. 7 shows the Pearson correlation coefficient between wind speed and precipitation for each grid cell in the domain, for the training and synthetic data, while Fig. 9 shows the tail dependence coefficient between wind speed and precipitation for each grid cell in the domain.

Table 2 summarises the correspondence of multi-hazard dependence metrics with the training data for two hazard combinations: wind with precipitation and wind with sea-level pressure. All metrics outperform the min-max scaling-only results, indicating that transforming to a well-defined distribution prior to training improves the dependence structure learning. The difference between the results for different training margins is small, with no single margin clearly outperforming the others across all metrics.

[Figure]

(a) Gaussian                      (b) Gumbel

(c) Uniform             (d) Min-max scaled (empirical only)

**Figure 7.** Pearson correlation between wind and precipitation for ERA5 (left) and synthetic (right) storm events

[Figure]

(a) Gaussian                      (b) Gumbel

(c) Uniform             (d) Min-max scaled (empirical only)

**Figure 8.** Pearson correlation between wind and precipitation for ERA5 (left) and synthetic (right) storm events

**Figure 9.** Tail dependence coefficient between wind and precipitation for ERA5 (left) and synthetic (right) storm events.

**Table 2.** Multi-hazard dependence metrics compared with training data.

| Variables | Margins | Correlation | | Tail dependence | |
|---|---|---|---|---|---|
| | | Pearson | MAE | Pearson | MAE |
| Wind–precipitation | Min-max scaled | 0.3756 | 0.1227 | 0.5504 | 0.1097 |
| | Uniform | **0.8641** | 0.0985 | 0.6197 | 0.1373 |
| | Gaussian | 0.8379 | 0.0863 | 0.6222 | 0.0925 |
| | Gumbel | 0.7792 | **0.0841** | **0.6311** | **0.0824** |
| Wind–pressure | Min-max scaled | 0.4330 | 0.0795 | 0.5464 | 0.0845 |
| | Uniform | **0.7667** | 0.0641 | **0.7131** | 0.0909 |
| | Gaussian | 0.7256 | **0.0628** | 0.6682 | 0.0878 |
| | Gumbel | 0.4915 | 0.0815 | 0.5274 | **0.0763** |

**References**

Bader, B., Yan, J., and Zhang, X.: Automated threshold selection for extreme value analysis via ordered goodness-of-fit tests with adjustment for false discovery rate, https://doi.org/10.1214/17-AOAS1092, 2018.

Bhatia, K., Vecchi, G., Murakami, H., Underwood, S., and Kossin, J.: Projected response of tropical cyclone intensity and intensification in a global climate model, Journal of climate, 31, 8281–8303, 2018.

Boulaguiem, Y., Zscheischler, J., Vignotto, E., van der Wiel, K., and Engelke, S.: Modeling and simulating spatial extremes by combining extreme value theory with generative adversarial networks, Environmental Data Science, 1, e5, 2022.

Coles, S., Bawa, J., Trenner, L., and Dorazio, P.: An Introduction to Statistical Modeling of Extreme Values, vol. 208, Springer, 2001.

Cook, N. J.: Reliability of Extreme Wind Speeds Predicted by Extreme-Value Analysis, Meteorology, 2, 344–367, https://doi.org/10.3390/meteorology2030021, 2023.

Cook, N. J. and Harris, R. I.: Exact and general FT1 penultimate distributions of extreme wind speeds drawn from tail-equivalent Weibull parents, Structural Safety, 26, 391–420, https://doi.org/10.1016/j.strusafe.2004.01.002, 2004.

Embrechts, P., McNeil, A., and Straumann, D.: Correlation and dependence in risk management: properties and pitfalls, Risk management: value at risk and beyond, 1, 176–223, 2002.

Harris, I.: Generalised Pareto methods for wind extremes. Useful tool or mathematical mirage?, Journal of Wind Engineering and Industrial Aerodynamics, 93, 341–360, 2005.

Harris, R. I.: XIMIS, a penultimate extreme value method suitable for all types of wind climate, Journal of Wind Engineering and Industrial Aerodynamics, 97, 271–286, 2009.

Harris, R. I. and Cook, N. J.: The parent wind speed distribution: Why Weibull?, Journal of wind engineering and industrial aerodynamics, 131, 72–87, 2014.

Healy, D., Tawn, J., Thorne, P., and Parnell, A.: Inference for extreme spatial temperature events in a changing climate with application to Ireland, Journal of the Royal Statistical Society Series C: Applied Statistics, 74, 275–299, 2025.

Heffernan, J. E. and Tawn, J. A.: A conditional approach for multivariate extreme values (with discussion), Journal of the Royal Statistical Society Series B: Statistical Methodology, 66, 497–546, 2004.

Huang, L., Antolini, F., Mostafavi, A., Blessing, R., Garcia, M., and Brody, S. D.: High-resolution flood probability mapping using generative machine learning with large-scale synthetic precipitation and inundation data, Computer-Aided Civil and Infrastructure Engineering, 2025.

Huser, R. and Wadsworth, J. L.: Advances in statistical modeling of spatial extremes, Wiley Interdisciplinary Reviews: Computational Statistics, 14, e1537, 2022.

Huster, T., Cohen, J., Lin, Z., Chan, K., Kamhoua, C., Leslie, N. O., Chiang, C.-Y. J., and Sekar, V.: Pareto GAN: Extending the representational power of GANs to heavy-tailed distributions, in: International Conference on Machine Learning, pp. 4523–4532, PMLR, 2021.

Keef, C., Tawn, J., and Svensson, C.: Spatial dependence in extreme river flows and precipitation in Great Britain, Journal of Hydrology, 378, 240–252, 2009.

Lamb, R., Keef, C., Tawn, J., Laeger, S., Meadowcroft, I., Surendran, S., Dunning, P., and Batstone, C.: A new method to assess the risk of local and widespread flooding on rivers and coasts, Journal of Flood Risk Management, 3, 323–336, 2010.

Quinn, N., Bates, P. D., Neal, J., Smith, A., Wing, O., Sampson, C., Smith, J., and Heffernan, J.: The spatial dependence of flood hazard and risk in the United States, Water Resources Research, 55, 1890–1911, 2019.

Reiss, R.-D. and Thomas, M.: Statistical Analysis of Extreme Values: With Applications to Insurance, Finance, Hydrology and Other Fields, Birkhäuser Verlag, Basel, 3 edn., ISBN 978-3-7643-7230-9, 2007.

Wiese, M., Knobloch, R., and Korn, R.: Copula & marginal flows: Disentangling the marginal from its joint, arXiv preprint arXiv:1907.03361, 2019.